# Area postrema neurons mediate interleukin-6 function in cancer cachexia

Qingtao Sun [1,6], Daniëlle van de Lisdonk [1,2,6], Miriam Ferrer [1,6], Bruno Gegenhuber[1], Melody Wu [1], Youngkyu Park [1], David A. Tuveson[1], Jessica Tollkuhn [1], Tobias Janowitz [1] & Bo Li [1,3,4,5] ✉

Interleukin-6 (IL-6) has been long considered a key player in cancer cachexia. It is believed that sustained elevation of IL-6 production during cancer progression causes brain dysfunctions, which ultimately result in cachexia. However, how peripheral IL-6 influences the brain remains poorly understood. Here we show that neurons in the area postrema (AP), a circumventricular structure in the hindbrain, is a critical mediator of IL-6 function in cancer cachexia in male mice. We find that circulating IL-6 can rapidly enter the AP and activate neurons in the AP and its associated network. Peripheral tumor, known to increase circulating IL-6, leads to elevated IL-6 in the AP, and causes potentiated excitatory synaptic transmission onto AP neurons and AP network hyperactivity. Remarkably, neutralization of IL-6 in the brain of tumor-bearing mice with an anti-IL-6 antibody attenuates cachexia and the hyperactivity in the AP network, and markedly prolongs lifespan. Furthermore, suppression of *Il6ra*, the gene encoding IL-6 receptor, specifically in AP neurons with CRISPR/ dCas9 interference achieves similar effects. Silencing Gfral-expressing AP neurons also attenuates cancer cachectic phenotypes and AP network hyperactivity. Our study identifies a central mechanism underlying the function of peripheral IL-6, which may serve as a target for treating cancer cachexia.

Cancer cachexia is a devastating metabolic wasting syndrome characterized by anorexia, fatigue, and dramatic involuntary bodyweight loss[1-4]. It affects 50-80% of cancer patients, lowering the quality of life, reducing tolerance to anticancer therapies, and drastically accelerating death[1,5]. The brain is known to have an important role in the pathogenesis of cancer cachexia[3,6,7]. In particular, recent studies implicate the hypothalamus, parabrachial nucleus, area postrema and other hindbrain structures in the development of cachectic phenotypes in animal models of cancer, such as anorexia, weight loss, and accelerated catabolic processes[8-14]. However, how the brain senses and reacts to peripheral cancers, thereby contributing to the development of cachectic phenotypes, is not well understood.

Possible mediators of cancer cachexia that may act as messengers to engage the brain during cancer progression include tumor-derived factors, metabolites from organs indirectly affected by tumor, and immune or inflammatory factors altered by tumor[2,3,6,7,15,16]. One such messenger is the pleiotropic cytokine IL-6[1-3,5,17-19]. Indeed, elevated levels of circulating IL-6 are associated with cancer progression and cachexia in patients and animal models[3,17,20-27]. Systemic administration of antibodies against IL-6 or IL-6 receptor shows anticachectic effects in human case reports[28-31]. Consistently, cancer cachexia in mouse models can be attenuated by peripherally administered antibodies against IL-6[32-37] or IL-6 receptor[38], or by deletion of the *Il6* gene[33,34]. These findings strongly indicate that IL-6 is a key mediator of cancer cachexia.

[1]Cold Spring Harbor Laboratory, Cold Spring Harbor, NY 11724, USA. [2]Center for Neuroscience, University of Amsterdam, Amsterdam, the Netherlands. [3]Westlake Laboratory of Life Sciences and Biomedicine, Hangzhou 310024 Zhejiang, China. [4]School of Life Sciences, Westlake University, Hangzhou 310024 Zhejiang, China. [5]Institute of Biology, Westlake Institute for Advanced Study, Hangzhou 310024 Zhejiang, China. [6]These authors contributed equally: Qingtao Sun, Daniëlle van de Lisdonk, Miriam Ferrer. ✉e-mail: bli@cshl.edu; libo@westlake.edu.cn

Most studies and therapeutic explorations on IL-6 in cancer cachexia has focused on its functions in peripheral organs, including the skeletal muscle, liver, and gut[17]. Although previous studies suggest that IL-6 may also influence brain functions – such as the regulation of food intake[39–41], fever[42] and the hypothalamic-pituitary-adrenal (HPA) axis[43] – how peripheral IL-6 is involved in these functions is unclear. In principle, IL-6 can activate its receptors on the terminals of peripheral nerves, which then transmit the signals to the brain[44]. Alternatively, circulating IL-6 may cross the blood-brain barrier (BBB) or reach circumventricular organs that lack or have a weak BBB, thereby acting within the brain[43,45,46].

In this work, we report that circulating IL-6 can rapidly enter the AP and activate AP neurons and their associated network. Peripheral tumor leads to elevated IL-6 in the AP, potentiated excitatory synaptic transmission onto AP neurons and hyperactivity in the AP network. Notably, neutralizing IL-6 in the brain or suppressing *Il6ra* in AP neurons attenuates cancer cachexia and AP network hyperactivity, and prolongs lifespan. Thus, AP neurons represent a critical mediator of IL-6 function in the development of cancer cachexia.

## Results

### The area postrema senses circulating IL-6

To determine if circulating IL-6 can enter the brain, we administered biotinylated IL-6 to the venous sinus of mice through retro-orbital injection (Fig. 1a; Methods). Considering that cancer cachexia is associated with chronic inflammation that may disrupt BBB integrity[47], we performed the experiment in both healthy mice and mice with cancer cachexia induced by the C26 adenocarcinoma (Fig. 1b; Methods). Mice in this model show a persistent increase in blood IL-6 levels, followed by robust cachectic phenotypes, including anorexia and dramatic bodyweight loss[3,17,25–27]. Three hours after the injection of biotinylated IL-6, we prepared brain sections from these mice in which the presence of the exogenous IL-6 was examined on the basis of avidin-biotin interactions. In the entire brain of both healthy mice and cachectic mice, we detected the peripherally administered IL-6 only in the area postrema (AP) (Fig. 1c, d; Supplementary Fig. 1), a circumventricular organ located outside of the BBB that has been implicated in nausea and vomiting response to emetic agents entering the circulation[48–51]. We did not detect the peripherally administered IL-6 in the median eminence (ME), which is another circumventricular organ[52,53] (Supplementary Fig. 1).

Immunohistochemistry in the AP revealed that intravenous IL-6 administration markedly increased the expression of Fos (Fig. 1e, f), an immediate early gene product linked to recent neuronal activation[54,55]. Interestingly, AP Fos expression was further increased in cachectic state (Fig. 1e, f). The systemic IL-6 administration also increased Fos expression in structures interconnected with the AP, including the nucleus tractus solitarii (NTS), parabrachial nucleus (PBN), paraventricular nucleus of the hypothalamus (PVN), central amygdala (CeA), bed nucleus of the stria terminalis (BNST), and arcuate hypothalamic nucleus (ARH) (Fig. 2). Fos expression in these structures did not further increase in cachectic state. Given these structures are interconnected[56–59] and receive monosynaptic or di-synaptic inputs from the AP[51,60], these results suggest that an increase in circulating IL-6 leads to increased activities in a network of brain areas encompassing the AP. The ME (which is adjacent to the ARH) and the dorsal part of the lateral septum (LS), on the other hand, showed no obvious IL-6-induced increase in Fos expression under either healthy or cachectic condition (Fig. 2).

Single molecule fluorescent in situ hybridization (smFISH) revealed that the *Il6ra*-expressing (*Il6ra*+) cells in the AP partially overlapped with *glucagon-like peptide 1 receptor*-expressing (*Glp1r*+) neurons (Fig. 1g–j; Supplementary Fig. 2), the major excitatory neuronal type in the AP[51]. About 17-18% of all the detected AP cells (which likely included glia cells) expressed both *Il6ra* and *Glp1r* (Fig. 1h & j). These *Il6ra*+ cells also partially overlapped with *Gfral*-expressing

(*Gfral*+) neurons (Fig. 1g, h; Supplementary Fig. 2), a subpopulation of *Glp1r*+ neurons in the AP that have been implicated in nausea and cancer cachexia[10,11,51]. The adjacent NTS also contains *Il6ra*+ cells and scattered *Gfral*+ neurons that do not express *Glp1r* (Supplementary Fig. 2). Interestingly, intravenous IL-6 administration induced *Fos* expression mainly in the *Il6ra*+ cells in the AP, and these cells partially overlapped with the *Glp1r*+ neurons (Fig. 1i, j). These results demonstrate that increased IL-6 in circulation is readily "sensed" by the AP, where it leads to neuronal activation that spreads to a network of interconnected areas within hours.

### Cancer causes AP hyperactivity

To investigate whether cancer, which is known to increase circulating IL-6[20–25], affects AP neurons, we again used mice inoculated with the C26 adenocarcinoma (Fig. 3a). We first measured IL-6 levels in the AP at different timepoints in this model. Notably, IL-6 was increased in the AP on day 7 following tumor inoculation, with the levels remaining elevated till the endpoint of the experiment when the animals had developed cachexia (Fig. 3b). In contrast, in the ME, IL-6 levels didn't increase until after the onset of cachexia, while in the cortex IL-6 levels didn't increase throughout the different stages (Supplementary Fig. 3). These results are consistent with the above observations that the AP is more accessible to circulating IL-6.

We then examined Fos expression in the brain in this model before the onset of cachexia (11 days after tumor inoculation). The number of Fos+ cells in the AP was markedly increased in the tumor-bearing mice compared with control mice (Fig. 3c, d). Interestingly, the NTS, PBN, PVN, BNST, and CeA – which are structures previously implicated in cancer cachexia[7,9–11] – also showed tumor-induced increase in Fos+ cells. These structures are the same as those showing IL-6-induced Fos expression (Fig. 1e, f; Fig. 2). As mentioned above, these structures are interconnected[51,56–60]. Thus, cancer progression and systemic IL-6 application lead to increased activities in a common AP network.

Next, we tested whether cachexia is associated with lasting functional changes in AP neurons. We prepared acute brain slices from healthy control mice as well as the tumor-bearing mice already showing cachectic phenotypes, and recorded synaptic responses from AP neurons (Fig. 3e). We found that the amplitude of miniature excitatory postsynaptic currents (EPSCs) was markedly increased, while the frequency of the EPSCs was unchanged in cachectic mice relative to control mice (Fig. 3f, g). In contrast, there was no difference in the inhibitory postsynaptic currents (IPSCs) between cachectic mice and control mice (Fig. 3h, i). These results demonstrate that cancer cachexia is accompanied by a potentiation of excitatory synaptic transmission onto AP neurons, which may lead to hyperactivity in these neurons.

### Neutralizing IL-6 in the brain attenuates cachexia and hyperactivity in the AP network

We reasoned that increased circulating IL-6 during cancer progression readily enters the AP and results in AP neuron activation – like the intravenously administered exogenous IL-6 (Fig. 1) – leading to hyperactivity in these neurons, which results in AP network hyperactivity and ultimately cachexia.

As a first step to test this hypothesis, we neutralized IL-6 in the brain by intracerebroventricular (i.c.v.) infusion of an antibody against IL-6, which was achieved using an implanted miniature pump (Methods). Continuous infusion of the anti-IL-6 antibody, or an isotype control antibody, was initiated at two different timepoints, with the first being at 10 or 12 days after tumor inoculation (Fig. 4a, b), a stage when the AP has increased IL-6 and AP neurons show elevated activity (Fig. 3b–d), but cachexia has not yet started. We reasoned that, if we were able to rescue the animals at this late stage of cancer progression, we would most likely also be able to protect the animals from getting

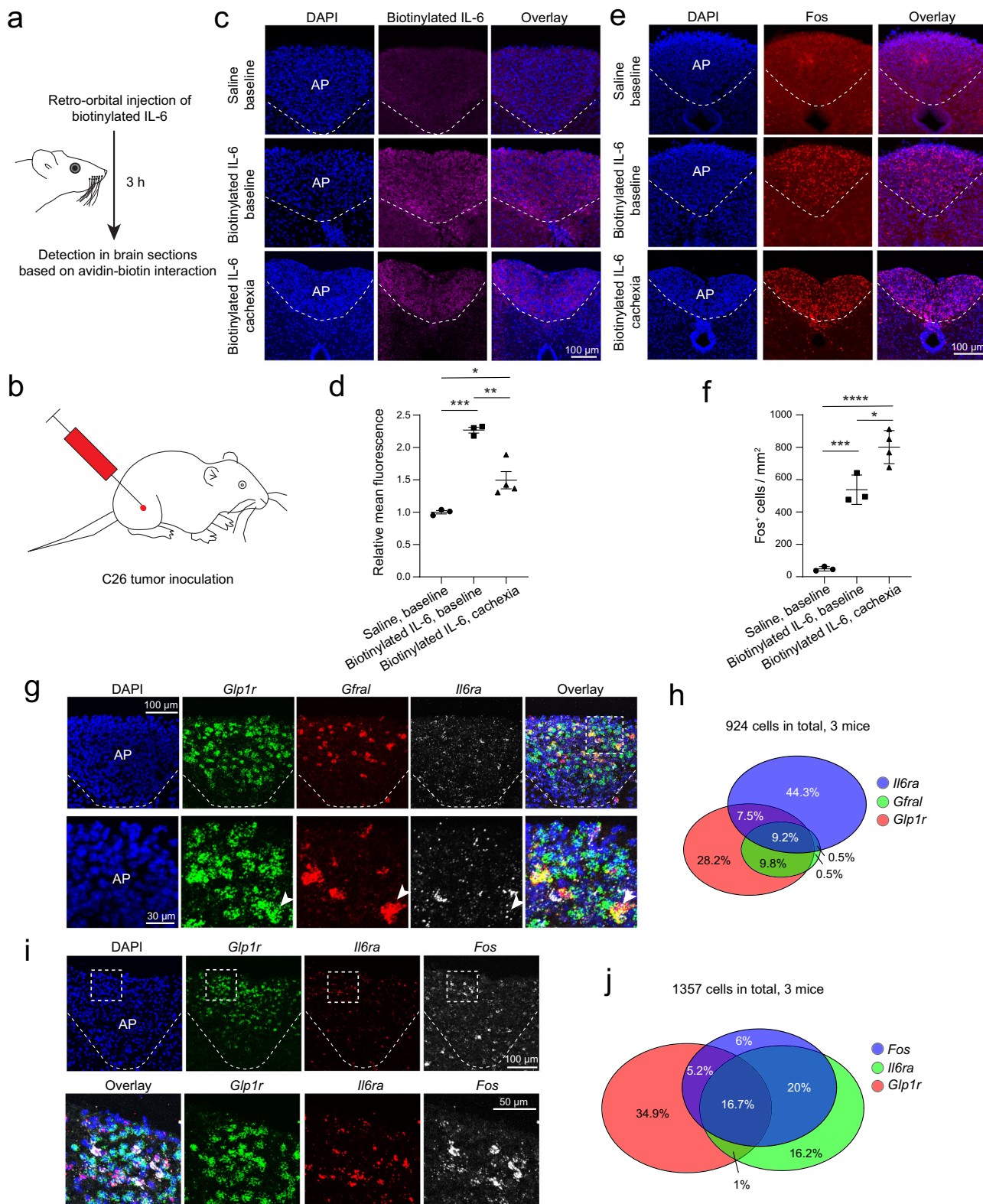

cachexia if the intervention were performed at earlier stages. Remarkably, compared with the control antibody, the anti-IL-6 antibody markedly reduced the cachectic phenotypes in almost all the mice, prolonging lifespans (Fig. 4c), reducing bodyweight loss and tissue loss (Fig. 4d, e; Supplementary Fig. 4a, b), increasing food and water intake (Fig. 4f), and increasing blood glucose levels (Supplementary Fig. 4c). Moreover, IL-6 antibody infusion reduced Fos expression in the AP, PBN, PVN, BNST, and, to a lesser extent, CeA

(Fig. 4g, h). The anti-IL-6 antibody did not change IL-6 levels in the plasma, but had a tendency to decrease IL-6 levels in the cerebrospinal fluid ($P = 0.067$; Supplementary Fig. 4d). As expected, the antibody did not stop tumor from growing (Supplementary Fig. 4e, f).

In the next experiment we initiated the i.c.v. infusion at an earlier timepoint, just before tumor inoculation (Fig. 5a). As expected, the pretreatment indeed also attenuated cachexia, with animals in the anti-IL-6 group showing reduced bodyweight loss, increased food intake,

**Fig. 1 | Circulating IL-6 can reach the area postrema (AP) and activate AP neurons. a, b** Schematics of the approach for retro-orbital injection of biotinylated IL-6 (**a**) and C26 tumor inoculation (**b**). **c** Confocal images showing the binding of the exogenous IL-6 to cells in the AP. **d** Quantification of the fluorescence signals from fluorescence-conjugated avidin in the AP, which recognizes the biotinylated exogenous IL-6 ($n$ = 3, 3, and 4 mice for saline baseline, IL-6 baseline, and IL-6 cachexia groups, respectively; F = 37.68, $P$ = 0.0002, *$P$ = 0.0212, ** $P$ = 0.002, ***$P$ = 0.001, one-way ANOVA followed by Tukey's multiple comparison test.). **e** Confocal immunohistochemical images showing Fos expression in the AP. **f** Quantification of Fos-expressing (Fos+) cells in the AP ($n$ = 3, 3, and 4 mice for saline baseline, IL-6 baseline, and IL-6 cachexia groups, respectively; F = 71.02, $P$ = 0.000022, *$P$ = 0.0103, ***$P$ = 0.00044, ****$P$ = 0.000017, one-way ANOVA

followed by Tukey's multiple comparison test). **g** Confocal images showing the expression of different genes in AP cells, detected with single molecule fluorescent in situ hybridization (smFISH). At the bottom are higher magnification images of the boxed area in the overlay image on the top. Arrowheads indicate a neuron that expresses all three genes. $n$ = 3 mice. **h** A Venn diagram showing the relationships among cells expressing *Il6ra*, *Gfral*, and *Glp1r* in the AP. **i** Characterization of the types of *Fos*+ cells in the AP by smFISH. At the bottom are higher magnification images of the boxed areas in images on the top. $n$ = 3 mice. **j** A Venn diagram showing the relationships among cells expressing *Fos*, *Il6ra*, and *Glp1r* in the AP. Data in d & f are presented as mean ± s.e.m. Source data are provided as a Source Data file.

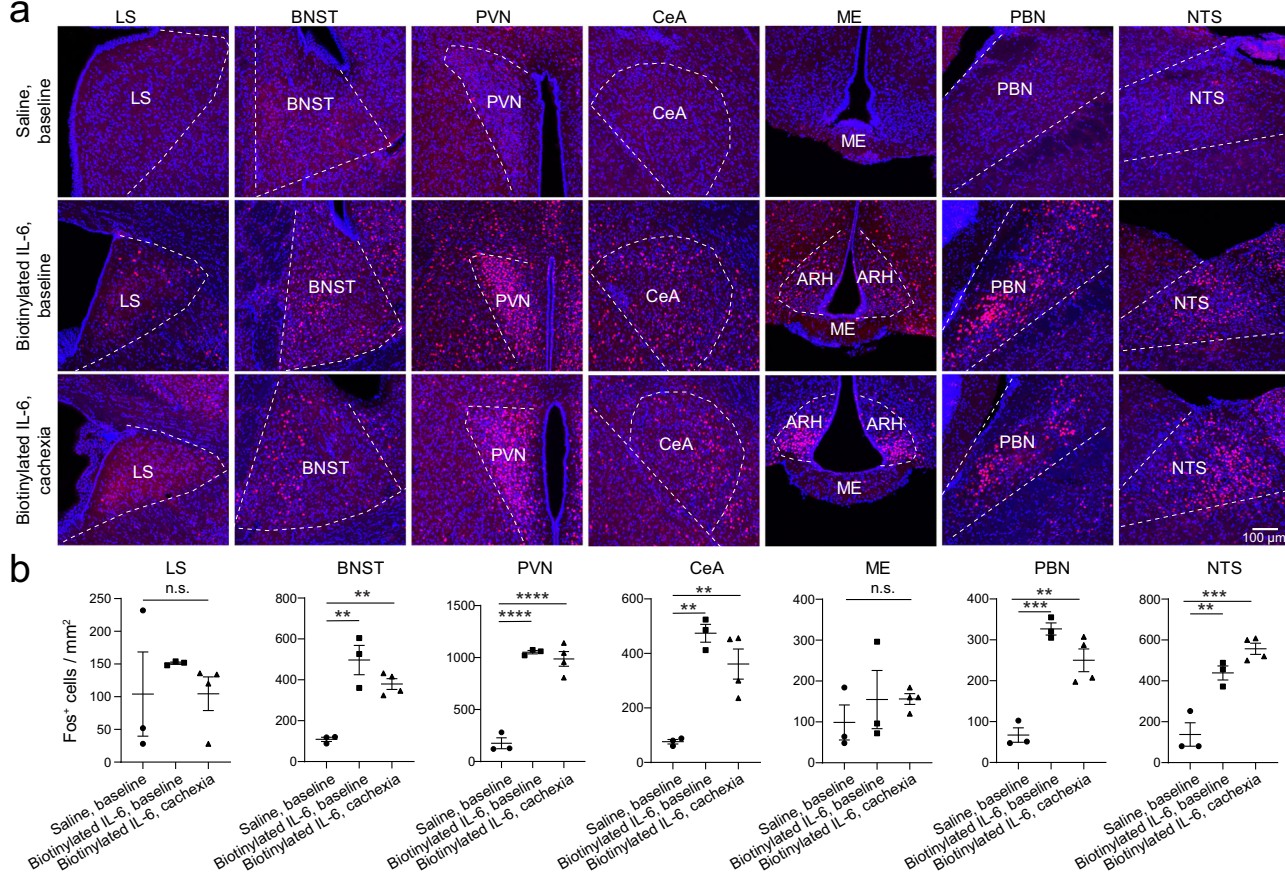

**Fig. 2 | Fos expression in different brain areas after retro-orbital injection of IL-6. a** Confocal immunohistochemical images showing Fos expression in different brain areas. **b** Quantification of Fos expression in different brain areas ($n$ = 3, 3, 4 mice in saline baseline, IL-6 baseline, and IL-6 cachexia groups, respectively; LS, F = 0.5, $P$ = 0.63 (n.s., nonsignificant); BNST, F = 21, $P$ = 0.0011, **$P$ = 0.001, **$P$ = 0.0055; PVN, F = 66.65, $P$ = 0.00003, ****$P$ = 0.00005, ****$P$ = 0.00005; CeA, F = 20.52, $P$ = 0.0012, **$P$ = 0.0011, **$P$ = 0.0052; ME, F = 0.55, $P$ = 0.6; PBN, F = 30.25,

$P$ = 0.0004, ***$P$ = 0.0003, **$P$ = 0.002; NTS, F = 30.15, $P$ = 0.0004, **$P$ = 0.0033, ***$P$ = 0.0003; one-way ANOVA followed by Tukey's multiple comparison test). LS, lateral septum; BNST, bed nucleus of the stria terminalis; PVN, paraventricular nucleus of hypothalamus; CeA, central amygdala; ME, median eminence; PBN, parabrachial nucleus; NTS, nucleus tractus solitarii. Data in b are presented as mean ± s.e.m. Source data are provided as a Source Data file.

and increased water intake compared with animals in the control group (Fig. 5b–d). The pretreatment also reduced tissue wasting, increased blood glucose levels, and prevented hyperactivity in the AP and its connected brain areas (Fig. 5e–g). Of note, unlike the experiment described above (Fig. 4 & Supplementary Fig. 4), in this experiment we euthanized both groups of mice at the same timepoint, therefore the two groups had similar tumor and spleen mass (Fig. 5e). Together, these experiments demonstrate that reducing IL-6 levels in the brain during cancer progression effectively attenuates cachexia, and also dampens the cancer-associated hyperactivity in the AP network.

## Suppressing *Il6ra* in AP neurons attenuates cancer cachexia

To investigate whether AP neurons mediate the functions of IL-6 in the development of cancer cachexia, we sought to suppress *Il6ra*, the gene encoding IL-6Rα, in these neurons using a recently developed CRISPR/dCas9 interference system. This system consists of dCas9-KRAB-MeCP2 [a fusion protein including the nuclease-dead Cas9 (dCas9), a Krüppel-associated box (KRAB) repressor domain, and the methyl-CpG binding protein 2 (MeCP2)] for transcriptional repression, and a CRISPR sgRNA (single guide RNA) for targeting the promoter region of genes of interest[61,62]. We designed and identified two sgRNAs, *Il6ra*-sgRNA-4 and −6, that resulted in suppression of *Il6ra* transcription

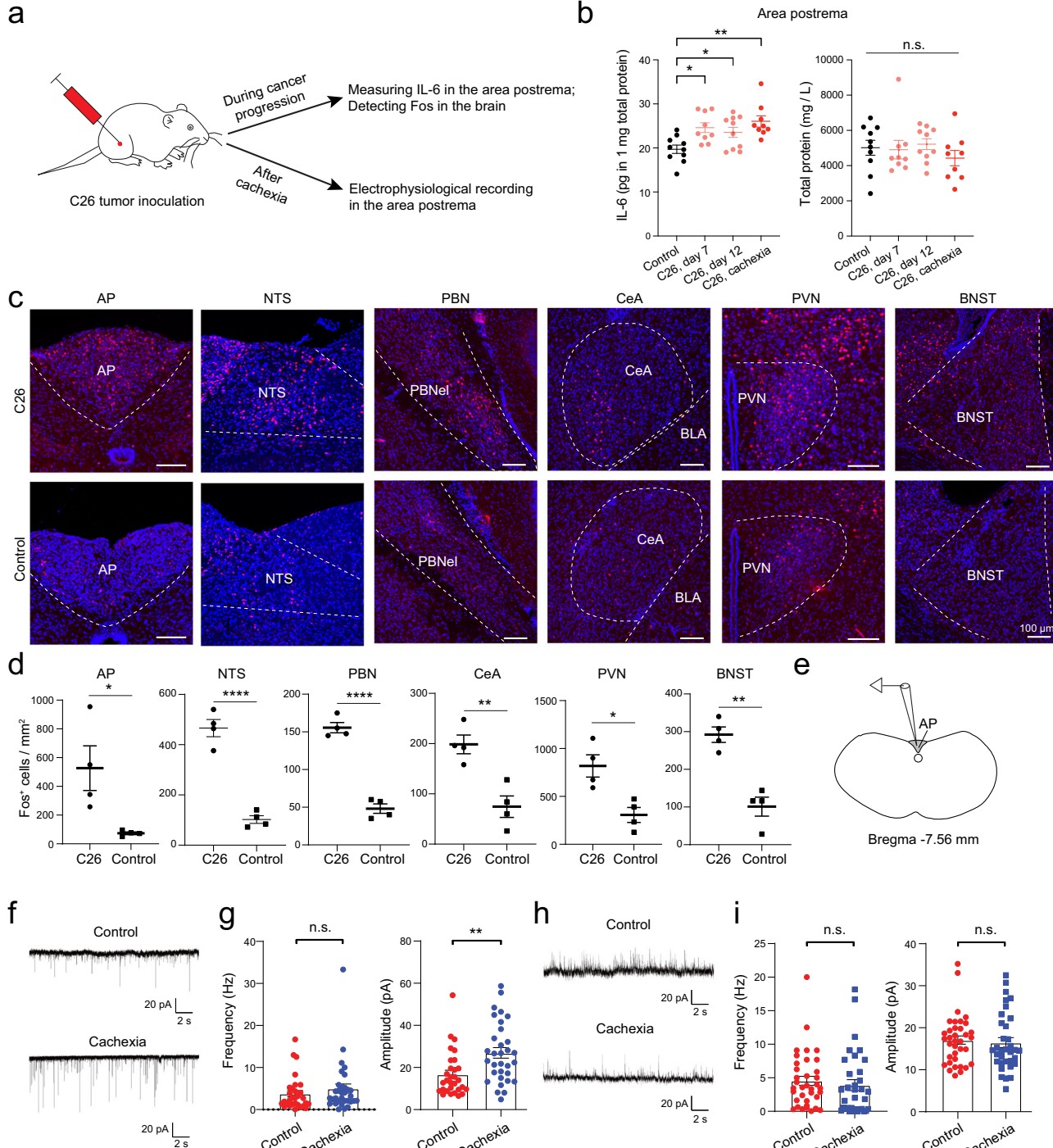

**Fig. 3 | C26 cancer causes increased IL-6 and neuron hyperactivity in the AP. a** A schematic of the experimental procedure. **b** IL-6 levels in the area postrema (AP) during cancer progression. IL-6 levels were normalized to total protein levels (n = 8-10 mice in each group; left, IL-6, F = 5.883, P = 0.0024, *P = 0.011, *P = 0.0483, **P = 0.001; right, total protein, F = 0.61, P = 0.61 (n.s., nonsignificant); one-way ANOVA followed by Tukey's post-hoc test). **c** Confocal immunohistochemical images showing Fos expression in different brain areas in tumor-bearing (top) or control (bottom) mice. **d** Quantification of Fos+ cells in different brain areas (n = 4 mice in each group; AP, t = 2.91, *P = 0.027, NTS, T = 9.77, ****P = 6.62×10⁻⁵, PBN, t = 11.65, ****P = 2.41 × 10⁻⁵, CeA, t = 4.37, **P = 0.0047, PVN, t = 3.65, *P = 0.011, BNST, t = 5.86, **P = 0.0011; unpaired t test). **e** A diagram showing the AP in a coronal brain section for electrophysiological recording. **f** Representative miniature EPSC traces

from AP neurons in control (top) and cachectic (bottom) mice. **g** Quantification of miniature EPSC frequency (left) and amplitude (right) (control, n = 30 cells / 6 mice, cachexia, n = 32 cells / 7 mice; frequency, n.s. (nonsignificant), P = 0.2513, amplitude, **P = 0.0017, Mann-Whitney test). **h** Representative spontaneous IPSC traces from AP neurons in control (top) and cachectic (bottom) mice. **i** Quantification of spontaneous IPSC frequency (left) and amplitude (right) (control, n = 36 cells / 7 mice, cachexia, n = 34 cells / 6 mice; frequency, n.s., P = 0.1637, amplitude, n.s., P = 0.4580, Mann-Whitney test). AP, area postrema; NTS, nucleus tractus solitarii; PBN, parabrachial nucleus; CeA, central amygdala; PVN, paraventricular nucleus of hypothalamus; BNST, bed nucleus of the stria terminalis. Data in b, d, g, i are presented as mean ± s.e.m. Source data are provided as a Source Data file.

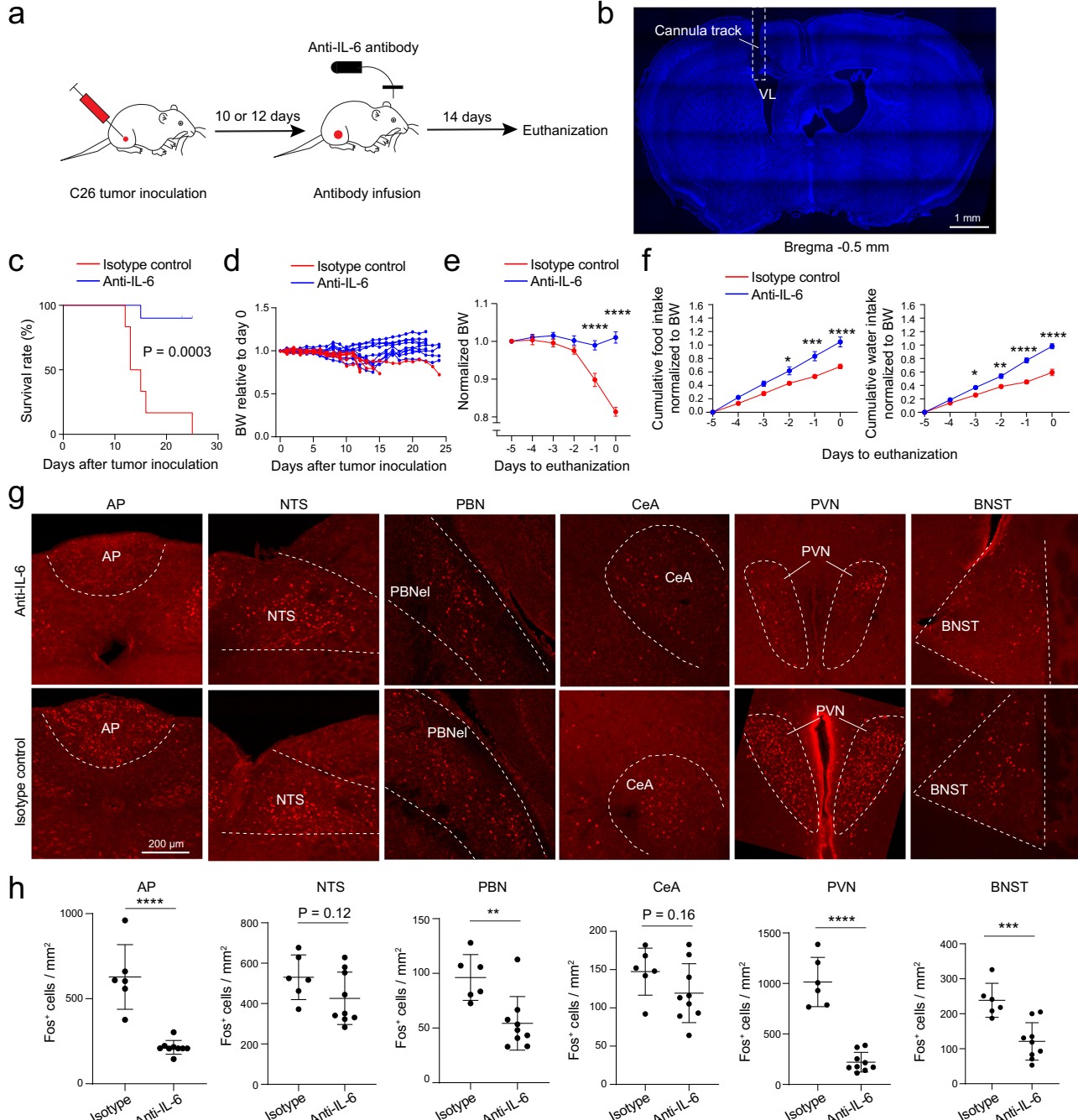

**Fig. 4 | Intracerebroventricular (i.c.v.) infusion of anti-IL-6 antibody before cachexia onset prevents cachexia in the C26 cancer model. a** A schematic of the experimental procedure. **b** A confocal image of a coronal brain section from a representative mouse, showing the location of the infusion cannula above the lateral ventricle (VL). **c** Survival curves of the mice after tumor inoculation (anti-IL-6 group, $n = 10$, isotype control group, $n = 6$; $P = 0.0003$, Mantel-Cox test).
**d** Bodyweight of individual mice relative to their bodyweight on the day of tumor inoculation. **e** Average bodyweight normalized to that on day -5 (anti-IL-6, $n = 10$, isotype control, $n = 6$; $F(1,84) = 75.13$, $P = 2.77 \times 10^{-13}$; day -1, ****$P = 1.2 \times 10^{-6}$, day 0, ****$P < 1 \times 10^{-15}$; two-way repeated-measures (RM) ANOVA followed by Sidak's post hoc test). f. Normalized cumulative food (left) and water (right) intake of the mice before being euthanized (anti-IL-6, $n = 10$, isotype control, $n = 6$ mice; food, $F(1,84) = 42.45$, $P = 5.08 \times 10^{-9}$, day -2, *$P = 0.045$, day -1, ***$P = 0.00018$, day 0,

****$P = 0.0000047$; water, $F(1,84) = 112.2$, $P < 1 \times 10^{-15}$, day -3, *$P = 0.034$, day -2, **$P = 0.0011$, day -1, ****$P = 2.17 \times 10^{-11}$, day 0, ****$P = 7 \times 10^{-15}$; two-way RM ANOVA with Sidak's post hoc test). g. Confocal immunohistochemical images showing Fos expression in different brain areas in the mice infused with the anti-IL-6 antibody (top) and the control antibody (bottom). h. Quantification of Fos+ cells in different brain areas (anti-IL-6, $n = 9$ mice, isotype control, $n = 6$ mice; AP, t = 8.11, ****$P = 1.83 \times 10^{-11}$, NTS, t = 1.62, $P = 0.12$, PBN, t = 0.8213, $P = 0.41$, CeA, t = 0.1375, $P = 0.89$, PVN, t = 15.55, ****$P < 10^{-15}$, BNST, t = 2.302, *$P = 0.0245$; t test with false discovery rate adjusted). AP, area postrema; NTS, nucleus tractus solitarii; PBN, parabrachial nucleus; CeA, central amygdala; PVN, paraventricular nucleus of hypothalamus; BNST, bed nucleus of the stria terminalis. Data in e, f, h are presented as mean ± s.e.m. Source data are provided as a Source Data file.

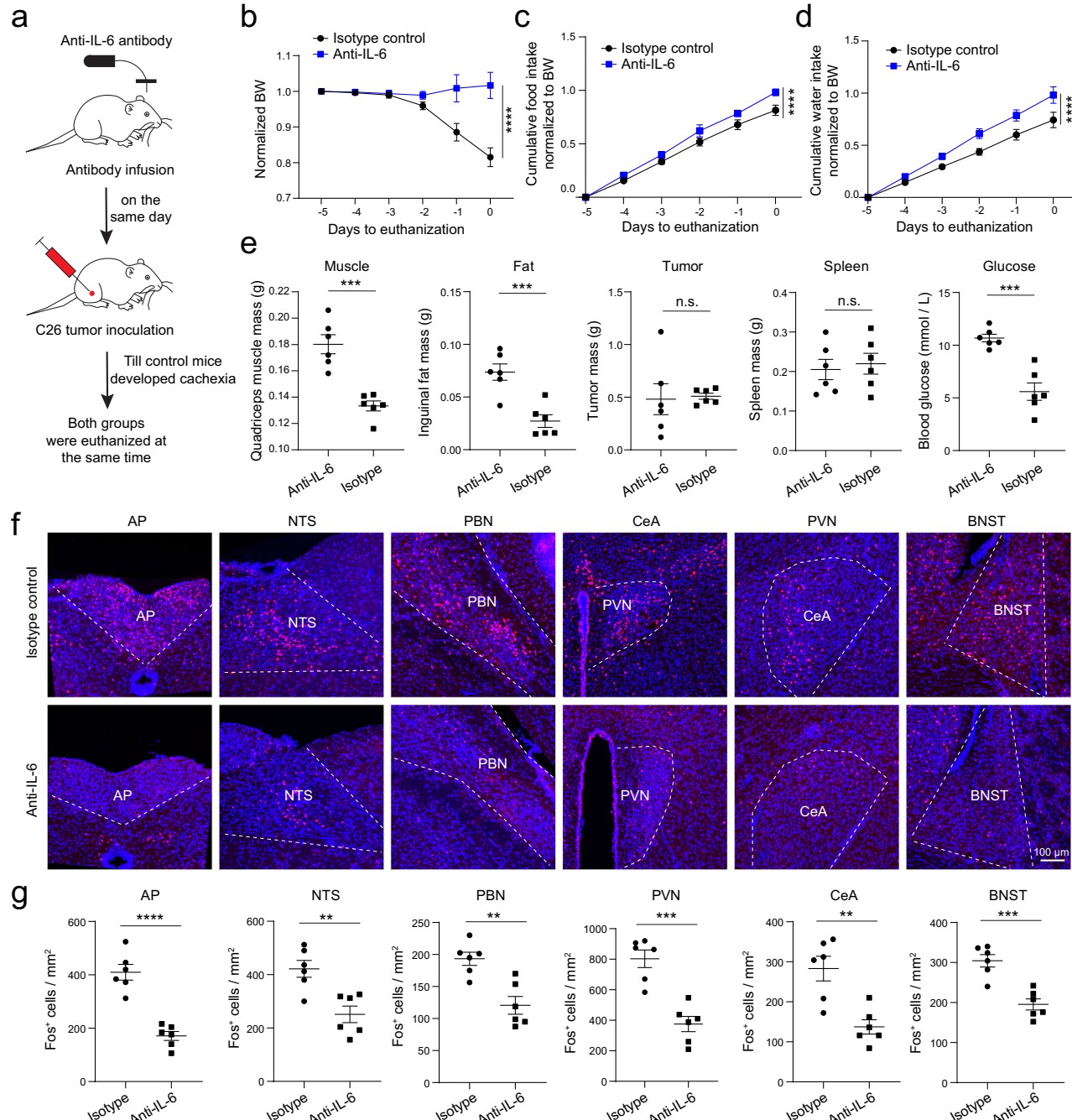

**Fig. 5 | Intracerebroventricular (i.c.v.) infusion of anti-IL-6 antibody before tumor inoculation prevents cachexia in the C26 cancer model. a** A schematic of the experimental procedure. **b** Average bodyweight (BW) normalized to that on day -5 (anti-IL-6, $n = 6$, isotype control, $n = 6$; $F_{(1,60)} = 28.46$, ****$P = 1.53 \times 10^{-6}$, two-way repeated-measures (RM) ANOVA followed by Sidak's post hoc test). **c** and **d** Normalized cumulative food (c) and water (d) intake of the mice before being euthanized ($n = 6$ animals in each group; c, $F_{(1,60)} = 19.31$, ****$P = 4.6 \times 10^{-5}$; d, $F_{(1,60)} = 26.06$, ****$P = 3.6 \times 10^{-6}$, two-way RM ANOVA followed by Sidak's post hoc test). **e** The effects of anti-IL-6 pretreatment ($n = 6$ mice in each group; muscle, $t = 5.72$, ***$P = 0.0002$; fat, $t = 4.74$, ***$P = 0.0008$; tumor, $t = 0.18$, $P = 0.858$ (n.s., nonsignificant); spleen, $t = 0.4$, $P = 0.7$ (n.s.); glucose, $t = 5.66$, ***$P = 0.0002$;

unpaired t test). **f** Confocal immunohistochemical images showing Fos expression in different brain areas in the mice infused with the isotype antibody (top) and the anti-IL-6 antibody (bottom). **g** Quantification of Fos[+] cells in different brain areas ($n = 6$ mice in each group, AP, $t = 7.03$, ****$P = 3.6 \times 10^{-5}$; NTS, $t = 3.87$, **$P = 0.003$; PBN, $t = 4.2$, **$P = 0.0018$; PVN, $t = 5.62$, ***$P = 0.0002$; CeA, $t = 4.08$, **$P = 0.0022$; BNST, $t = 5.21$, ***$P = 0.0004$; unpaired t test.). AP, area postrema; NTS, nucleus tractus solitarii; PBN, parabrachial nucleus; CeA, central amygdala; PVN, paraventricular nucleus of hypothalamus; BNST, bed nucleus of the stria terminalis. Data in **b**–**e**, **g** are presented as mean ± s.e.m. Source data are provided as a Source Data file.

when co-expressed with dCas9-KRAB-MeCP2 in in vitro screen (Supplementary Fig. 5).

We injected the AP with a lentivirus expressing dCas9-KRAB-MeCP2 under a neuronal promoter (lenti-SYN-FLAG-dCas9-KRAB-MeCP2), together with a lentivirus expressing *Il6ra*-sgRNA-4 (lenti-U6-

*Il6ra* sgRNA-4/EF1α-mCherry), or a sgRNA targeting the bacterial gene *lacZ* as a control (lenti-U6-*lacZ* sgRNA/EF1α-mCherry)[61] (Supplementary Fig. 6). Three weeks later, we prepared brain sections containing the AP from these mice for immunohistochemistry, which revealed that most of the FLAG signals colocalized with NeuN signals

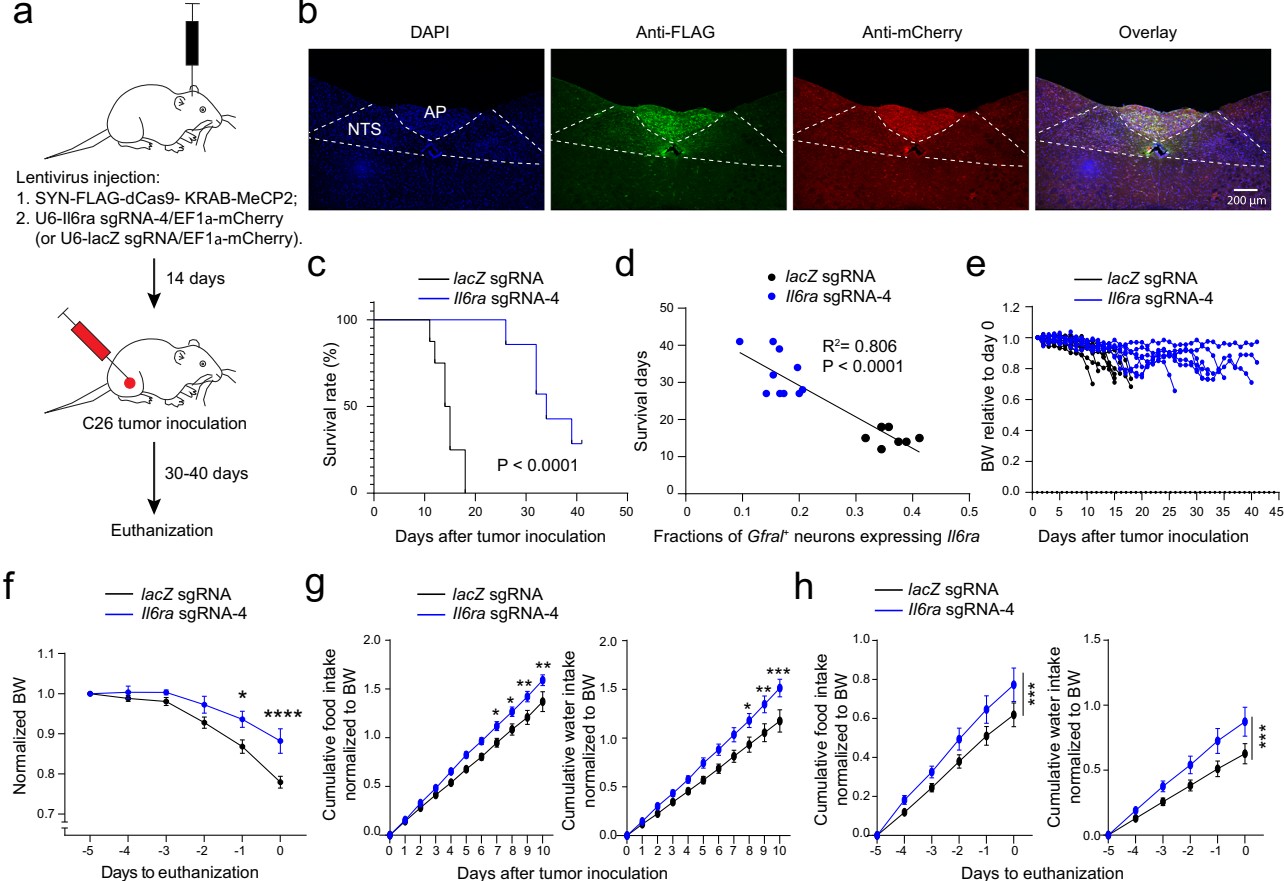

**Fig. 6 | Suppression of *Il6ra* expression in AP neurons attenuates cachexia in the C26 cancer model. a** A schematic of the experimental procedure. **b** Confocal immunohistochemical images of a coronal brain section from a representative mouse, showing the infection of AP cells with lentiviruses expressing the sgRNA (tagged with mCherry) and dCas9-KRAB-MeCP2 (tagged with FLAG). mCherry and FLAG were recognized by antibodies. The yellow cells in the overlay image indicate the dual-color labeled cells. *lacZ* sgRNA, *n* = 8, *Il6ra* sgRNA-4, *n* = 7. **c** Survival curves of the mice after tumor inoculation (*lacZ* sgRNA, *n* = 8, *Il6ra* sgRNA-4, *n* = 7; *P* = 0.000013, Mantel-Cox test). **d** Relationship between survival days and the fractions of *Gfral*⁺ neurons expressing *Il6ra* (*n* = 17 mice, R² = 0.806, F test, F = 62.34, *P* = 10⁻⁶ by a linear regression). **e** Bodyweight of individual mice relative to their bodyweight on the day of tumor inoculation. **f** Average bodyweight normalized to that on day -5 (*lacZ* sgRNA, *n* = 8, *Il6ra* sgRNA-4, *n* = 7; F(1,78) = 23.63, *P* = 9.6 × 10⁻⁶,

*P* = 0.012, ****P* = 0.00004, two-way repeated-measures (RM) ANOVA with Sidak's post hoc test). **g** Normalized cumulative food (left) and water (right) intake of the mice after tumor inoculation (*lacZ* sgRNA, *n* = 8, *Il6ra* sgRNA-4, *n* = 7; food, F(1130) = 46.9, *P* = 2.04 × 10⁻¹⁰; day 7, *P* = 0.046, day 8, *P* = 0.026, day 9, **P* = 0.0046, day 10, **P* = 0.003; water, F(1130) = 43.87, *P* = 6.6 × 10⁻¹⁰; day 8, *P* = 0.03, day 9, **P* = 0.005, day 10, ***P* = 0.0006; two-way RM ANOVA with Sidak's post hoc test). **h** Normalized cumulative food (left) and water (right) intake of the mice in the 5 days before being euthanized (*lacZ* sgRNA, *n* = 8, *Il6ra* sgRNA-4, *n* = 7; food, F(1,65) = 12.82, ***P* = 0.0006; water, F(1,65) = 16.37, ***P* = 0.00012; two-way RM ANOVA with Sidak's post hoc test). AP, area postrema; NTS, nucleus tractus solitarii. Data in f, g, h are presented as mean ± s.e.m. Source data are provided as a Source Data file.

(Supplementary Fig. 6a, b), indicating that viral expression was specific to neurons. Furthermore, smFISH showed that the *Il6ra*-sgRNA-4 group had reduced *Il6ra* expression in AP neurons, including the *Gfral*⁺ neurons (Supplementary Fig. 6c, d), showing the efficiency of this approach.

We then examined the effects of this approach on cachexia. We injected the knock-down or control viruses into the AP and, two weeks later, inoculated the mice with the C26 tumor (Fig. 6a, b). Notably, the *Il6ra*-sgRNA-4 group had markedly increased lifespans (Fig. 6c), an effect that was inversely correlated with the percentage of *Gfral*⁺ neurons that had detectable *Il6ra* expression (Fig. 6d). The *Il6ra*-sgRNA-4 group also had reduced bodyweight loss (Fig. 6e, f), increased food and water intake (Fig. 6g, h), increased blood glucose levels (Supplementary Fig. 7a), and had a tendency to reduce tissue loss (Supplementary Fig. 7b, c). At the endpoint of the experiment, mice in the *Il6ra*-sgRNA-4 group had larger tumor and spleen compared with mice in the control group (Supplementary Fig. 7d, e), presumably because of the increase in lifespan in the former group.

In a separate experiment, we euthanized the *Il6ra*-sgRNA-4 mice and the *lacZ*-sgRNA control mice on the same day, as soon as the latter had developed cachexia (Fig. 7a–j). This design ensured that the two groups had tumor for the same duration and thus had comparable tumor and spleen mass (Fig. 7g, h). Consistent with the above observations, the *Il6ra*-sgRNA-4 group had reduced bodyweight loss and tissue loss (Fig. 7b–d), even though these mice had similar food and water intake to the control mice before the termination of this experiment (Fig. 7e, f). Notably, the *Il6ra*-sgRNA-4 mice had reduced Fos expression compared with the *lacZ* sgRNA mice in the AP, PBN, and PVN (Fig. 7i, j), confirming that suppressing *Il6ra* expression in AP neurons lowers the cancer-associated hyperactivity in the AP network.

Additional experiments showed that the other sgRNA, the *Il6ra* sgRNA-6, had similar effects to those of *Il6ra* sgRNA-4 (Supplementary Fig. 8). Together, these results indicate that the IL-6Rα on AP neurons, especially that on *Gfral*⁺ neurons, is a critical mediator of IL-6 function in the development of cancer cachexia in the C26 model.

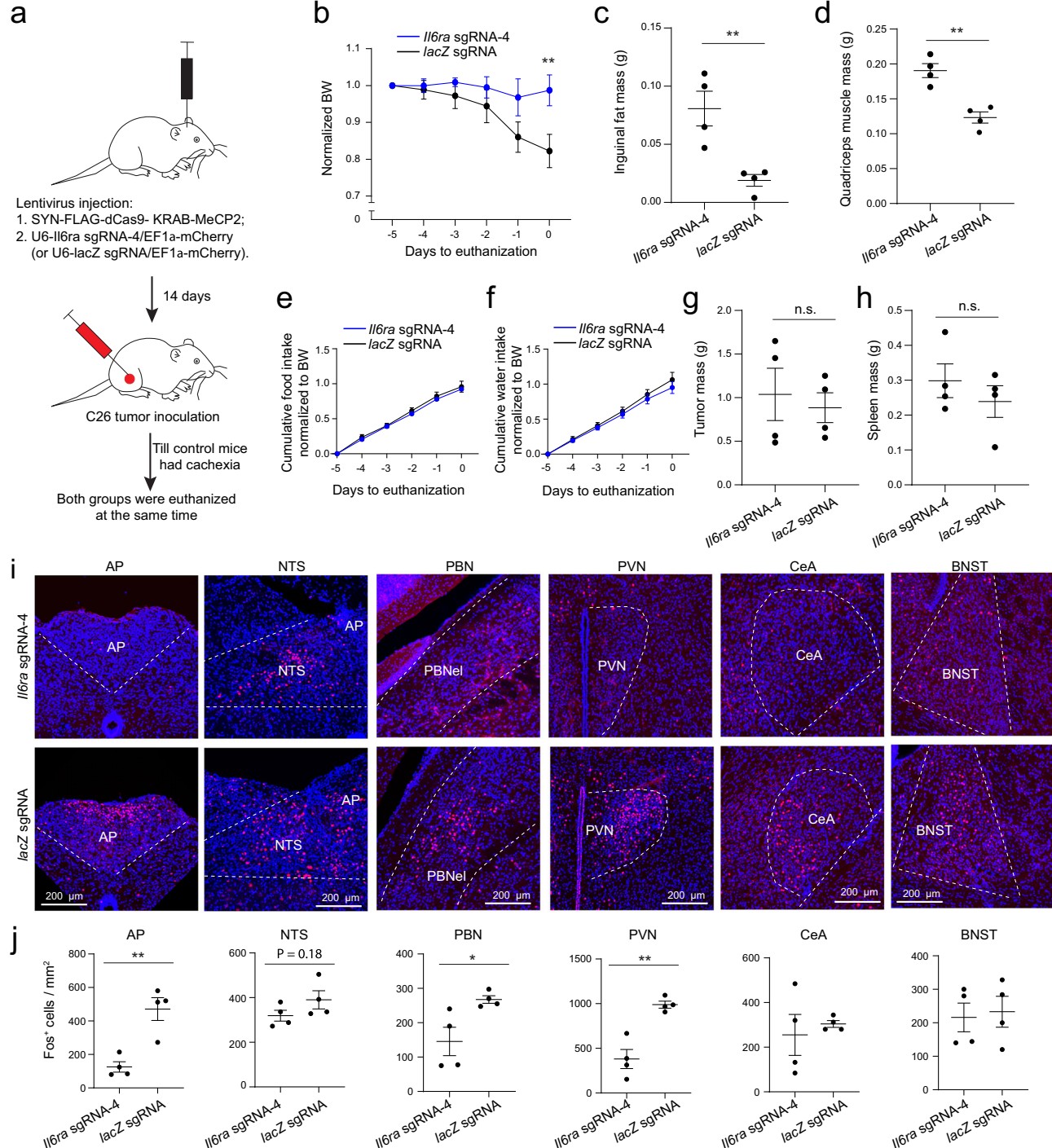

**Fig. 7 | Suppression of *Il6ra* expression in AP neurons attenuates cachexia and reduces the hyperactivity in the AP network in the C26 cancer model. a** A schematic of the experimental procedure. When one animal in the *lacZ* sgRNA (control) group became cachectic, that animal and a randomly selected animal in the *Il6ra* sgRNA-4 group were euthanized to check Fos expression and other phenotypes. **b** Average bodyweight normalized to that on day -5 (F(1,36) = 10.45, P = 0.0026, **P = 0.0072, two-way repeated-measures (RM) ANOVA with Sidak's post hoc test, *lacZ* sgRNA group, n = 4 mice, *Il6ra* sgRNA-4 group, n = 4 mice). **c** Inguinal fat mass (t = 3.941, **P = 0.0079, t test). **d** Quadriceps muscle mass (t = 5.26, **P = 0.0019, t test). **e, f** Normalized cumulative food (**e**) and water (**f**) intake of the mice before being euthanized (food, F(1, 36) = 1.83, P = 0.18; water, F(1,

36) = 1.824, P = 0.19; two-way RM ANOVA with Sidak's post hoc test). **g, h** Tumor (**g**) and spleen (**h**) mass of the mice (tumor, t = 0.444, P = 0.6725; spleen, t = 0.898, P = 0.404; t test). **i** Confocal immunochemical images showing Fos expression in different brain areas in representative mice of the two groups. **j** Quantification of Fos⁺ cells in different brain areas (AP, t = 4.617, **P = 0.0036; NTS, t = 1.5, P = 0.18; PBN, t = 2.849, *P = 0.029; PVN, t = 5.332, **P = 0.0018; CeA, t = 0.5267, P = 0.617; BNST, t = 0.27, P = 0.796; t test with false discovery rate adjusted). AP, area postrema; NTS, nucleus tractus solitarii; PBN, parabrachial nucleus; PVN, paraventricular nucleus of hypothalamus; CeA, central amygdala; BNST, bed nucleus of the stria terminalis. Data in b-h & j are presented as mean ± s.e.m. Source data are provided as a Source Data file.

## Suppressing *Il6ra* in AP neurons attenuates cachexia in a PDAC model

We further tested whether suppressing *Il6ra* in AP neurons could prevent cachexia in an orthotopic tumor model, which is a well characterized pancreatic ductal adenocarcinoma (PDAC) model based on the FC1245 clonogenic cell line[63,64] (Methods). Mice in this model showed decreased food and water intake, and reduced muscle weight and fat weight (Supplementary Fig. 9a-d), which are consistent with findings from previous studies[63,64]. These mice showed no bodyweight reduction (Supplementary Fig. 9c), which is also consistent with previous findings that the abdominal ascites and third spacing edema mask bodyweight reduction in this model[63,64]. The tumor bearing animals also showed other features of cachexia, including increased IL-6 levels in the plasma (Supplementary Fig. 9d) and increased Fos expression in the AP network (Supplementary Fig. 9e, f).

As described above, we again delivered the CRISPR/dCas9 system containing the *Il6ra*-sgRNA-4 or the control *lacZ*-sgRNA to AP neurons (Fig. 8a). Two weeks after the viral injection, we inoculated FC1245 cells in the pancreas of these mice (Methods). We found that the *Il6ra*-sgRNA-4 mice had improved food and water intake, and decreased tissue wasting compared with the control mice (Fig. 8b, c). The *Il6ra*-sgRNA-4 mice also had reduced Fos expression in the AP and its interconnected areas (Fig. 8d, e). Thus, suppressing *Il6ra* in AP neurons attenuates cachexia and AP network hyperactivity in the PDAC model.

## Inhibition of Gfral+ neurons in the AP attenuates anorexia in the LLC model

Given the result that the survival of mice in the C26 model was correlated with the suppression of *Il6ra* in *Gfral*+ neurons in the AP (Fig. 6d), and previous findings that Gfral and its ligand GDF-15 (Growth/differentiation factor 15) are involved in cancer cachexia[10,11], we reasoned that the activity of *Gfral*+ AP neurons contributes to the development of this syndrome. To test this hypothesis, we sought to selectively manipulate these neurons using the *Gfral-p2a-Cre* mice[51] in combination with a Cre-dependent viral approach. As the C26 model necessitates the use of Balb/c or CD2F1 mice[65], it is incompatible with the *Gfral-p2a-Cre* mice which have a C57BL/6 genetic background. Therefore, for this experiment, we used the implantable Lewis lung carcinoma (LLC) model, another established murine cancer model exhibiting features of cachexia, albeit milder than those of the C26 model[3,9,17].

We first measured the levels of IL-6 and GDF-15 in the blood at different timepoints of cancer progression in this model. Plasma IL-6 and GDF-15 were both increased at around two weeks following tumor inoculation (Fig. 9a). To specifically inhibit the activity of *Gfral*+ neurons, we injected the AP of *Gfral-p2a-Cre* mice bilaterally with an adeno-associated virus (AAV) expressing the tetanus toxin light chain (TeLC, which blocks neurotransmitter release[66]), or GFP (as a control) in a Cre-dependent manner. Two weeks later, these mice were inoculated with the LLC (Fig. 9b, c). Notably, compared with the GFP mice with tumor, the TeLC mice with tumor exhibited increased food intake at the late stage of cancer progression (Fig. 9d) and increased muscle mass at the endpoint (Fig. 9e), to levels comparable to the GFP sham group (i.e., mice that expressed the virally delivered GFP in *Gfral*+ neurons and received saline injection instead of tumor inoculation). The TeLC tumor mice also had increased fat and muscle mass compared with the GFP tumor mice (Fig. 9e). Moreover, the former group showed reduced Fos expression than the latter group in the AP, PBN, PVN, CeA and BNST (Fig. 10a, b). The two groups had similar tumor and spleen mass (Fig. 10c). These results indicate that reducing *Gfral*+ AP neuron activity attenuates the cachectic phenotypes. In addition, other areas in the AP network are likely also involved in this process.

## Discussion

Our study identifies AP neurons as a critical mediator of the function of IL-6 that leads to cancer cachexia in mice. We show that, first, circulating IL-6 rapidly enters the AP and causes the activation of neurons in the AP and its interconnected areas within hours. Second, peripheral tumors result in increased IL-6 levels in the AP and enhanced excitatory synaptic drive onto AP neurons, leading to neuronal hyperactivity in the AP network. Third, neutralization of IL-6 in the brain of tumor-bearing mice, via i.c.v. infusion of an IL-6 antibody, attenuates cachexia, counteracts the cachexia-associated hyperactivity in the AP network, and markedly prolongs lifespan. Fourth, specific suppression of *Il6ra* in AP neurons with CRISPR/dCas9 interference achieves similar effects. Lastly, specific suppression of the activities of *Gfral*+ neurons, which partially overlap with *Il6ra*+ cells in the AP (Fig. 1g, h; Supplementary Fig. 6c, d) and are involved in the effects of *Il6ra* suppression (Fig. 6d), also attenuates cancer-associated cachectic phenotypes and AP network hyperactivity.

We utilized three cachexia models: C26, LLC and PDAC. These models have shared features related to cachexia, including tissue wasting, anorexia, increased IL-6 production, and hyperactivity in the AP network. However, they also have differences. For example, body-weight loss is only apparent in the C26 model but not in the other two models. In the PDAC model, the abdominal ascites and third spacing edema may mask bodyweight reduction[63,64] (also see Fig. 8b left; Supplementary Fig. 9c). The C26 model also has the highest IL-6 levels in the plasma that may, at least in part, be responsible for the more severe phenotypes in this model. Despite the differences, the cachectic phenotypes in these models can all be attenuated by suppressing the IL-6 signaling or activity of AP neurons.

The AP network – including the AP, NTS, PBN, PVN, BNST, and CeA, which are interconnected[51,56–60] – show robust hyperactivity during cancer progression, starting from the pre-cachectic state when IL-6 starts to elevate in the AP (Fig. 3b-d) till cachectic state in different tumor models (Fig. 4g, h; Fig. 5f, g; Fig. 7i, j; Fig. 8d, e; and Fig. 10a, b). Interestingly, intravenous injection of IL-6 also induces the same pattern of hyperactivity in this network (Fig. 1e, f; Fig. 2). This network has also been shown to be hyperactive during inflammation[67,68]. These findings support the idea that the AP network can detect peripheral inflammatory processes (including those associated with cancer) that involve IL-6.

The AP network has been implicated in regulating feeding behavior and metabolism[51,59,69–76]. In particular, the AP and the neighboring NTS, as well as the PBN and the PVN have previously been implicated in cancer cachexia[7,9–11]. The AP sends direct projections to the PBN and the NTS, and the NTS also directly projects to the PBN as well as the PVN[51,60,77]. Given previous findings that both the PBN[9,51,58,59,77–79] and the PVN[6,80,81] are involved in feeding suppression, it is possible that the AP drives cancer-associated anorexia via the AP→PBN, the AP→NTS→PBN, or the AP→NTS→PVN pathway.

A notable observation is that the AP also drives weight loss independent of anorexia during cancer progression (Fig. 7), consistent with findings that cancer cachexia involves active catabolic processes in addition to anorexia, and the tissue wasting can only be partially reversed by nutritional support[1,3,5]. Interestingly, multiple nuclei in the AP network are connected with mechanisms that promote catabolic process in peripheral organs[69,70,73,76,82], providing an anatomical basis for this function of the AP.

Recent studies indicate that Gfral is exclusively expressed by neurons in the AP and the NTS[83–86], and systemic administration of GDF-15 activates GFRAL+ neurons in the AP and induces vomiting and anorexia[10,51,87–90]. Furthermore, neutralization of Gfral or GDF-15 with antibodies attenuates cancer-associated cachectic phenotypes in animals[10,11]. Thus, GDF-15 may also influence cancer cachexia, like IL-6, through the AP network. However, as GDF-15 functions as a central alert to the organism in response to a broad range of stressors[91], including infection, blockade of GDF-15/GFRAL is likely to have detrimental effects if used as a therapeutic strategy. Indeed, it has recently been shown that GDF-15 is essential for surviving bacterial and viral infections[72].

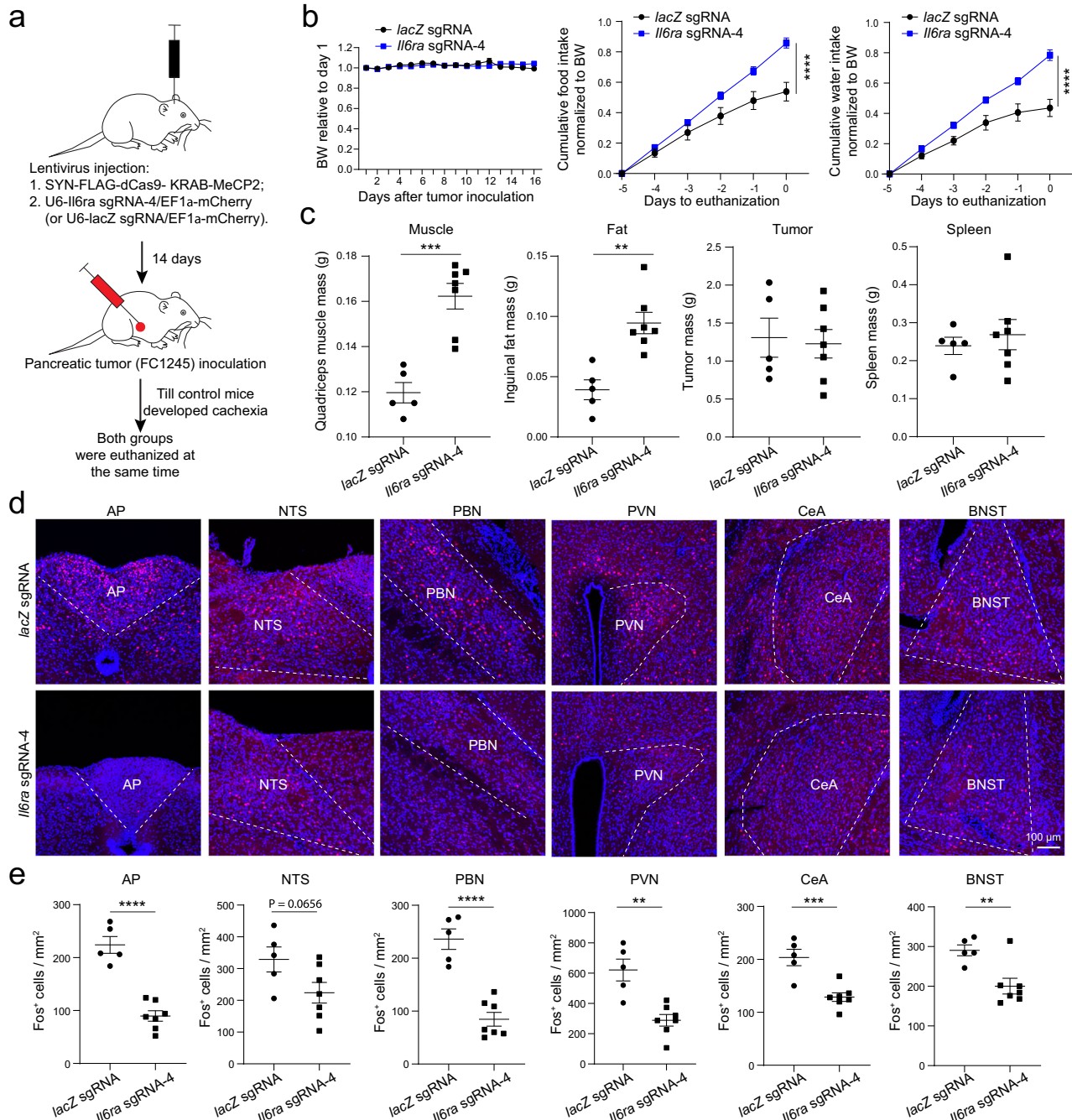

**Fig. 8 | Suppression of *Il6ra* expression in AP neurons attenuates cachexia in the pancreatic cancer model. a** A schematic of the experimental procedure.
**b** Normalized bodyweight (BW; left), food intake (middle), and water intake (right). Bodyweight was normalized to the initial bodyweight when tumor was implanted. Bodyweight, F(1,130) = 0.0021, *P* = 0.96. Food intake, F(1,60) = 41.91, ****P* = 1.96 × 10⁻⁸. Water intake, F(1,60) = 69.55, ****P* = 1.3 × 10⁻¹¹. *n* = 5 and 7 mice in *lacZ* sgRNA (control) group and *Il6ra* sgRNA-4 group, respectively. Two-way RM ANOVA with Sidak's post hoc test. **c** The effects of suppressing *Il6ra* expression (*n* = 5 and 7 mice in *lacZ* sgRNA (control) group and *Il6ra* sgRNA-4 group, respectively; muscle, t = 5.51, ****P* = 0.0003; fat, t = 4.37, ***P* = 0.0014; tumor, t = 0.26, *P* = 0.8014; spleen,

t = 0.574, *P* = 0.5786, unpaired t test). **d** Confocal immunohistochemical images showing Fos expression in different brain areas in control group (top) and in *Il6ra* knock-down group (bottom). **e** Quantification of Fos expression in different brain areas (*n* = 5 and 7 mice in *lacZ* sgRNA (control) group and *Il6ra* sgRNA-4 group, respectively; AP, t = 7.58, ****P* = 1.88 × 10⁻⁵; NTS, t = 2.07, *P* = 0.0656; PBN, t = 6.78, ****P* = 4.84 × 10⁻⁵; PVN, t = 4.4, ***P* = 0.0013; CeA, t = 4.64, ****P* = 0.0009; BNST, t = 3.41, ***P* = 0.0066. Unpaired t test). AP, area postrema; NTS, nucleus tractus solitarii; PBN, parabrachial nucleus; PVN, paraventricular nucleus of hypothalamus; CeA, central amygdala; BNST, bed nucleus of the stria terminalis. Data in **b**, **c**, **e** are presented as mean ± s.e.m. Source data are provided as a Source Data file.

IL-6 has long been known as a key contributor to cancer cachexia[1–3,5,17–19]. Efforts exploring IL-6 as a potential therapeutic target thus far have been focused on peripheral IL-6 or IL-6 receptors, and relied on systemic application of antibodies against these molecules[3,17,31]. However, such systemic approach may not be effective in reducing IL-6 signaling in the brain. Furthermore, as IL-6 is a pleiotropic cytokine essential for immune and metabolic functions, with receptors widely distributed in the entire organism[17,92], systemic neutralization of IL-6 or its receptors will compromise these functions globally and likely cause severe side effects[93,94]. Our results from multiple cancer models suggest that targeting IL-6 signaling in the brain, or more specifically in the AP, could be an effective avenue for treating cancer cachexia.

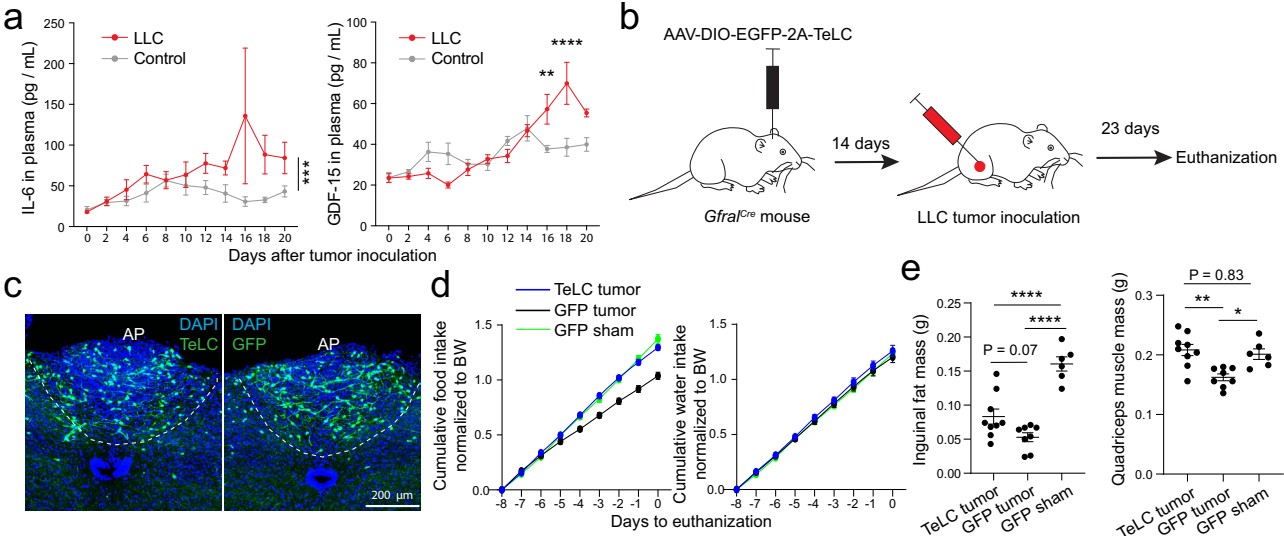

**Fig. 9 | Inhibition of *Gfral*⁺ AP neurons attenuates cachexia in the Lewis lung cancer (LLC) model. a** Plasma IL-6 (left) and GDF-15 (right) concentrations (IL-6: control, 7 mice except day 2 and 8, where there are 5 mice; LLC, 8 mice except day 4 where there are 6 mice, day 6 and 12 where there are 11 mice, and day 10 and 18 where there are 7 mice; F(1128) = 14.12, ***P = 0.0003, two-way repeated-measures ANOVA with Sidak's post hoc test; GDF-15: control, 5 mice except day 0 and 14 where there are 8 and 3 mice, respectively; LLC, 5 mice except day 0 where there are 7 mice; F(1,91) = 2.571, P = 0.11, **P = 0.007, ****P = 1.74 × 10⁻⁶, two-way repeated-measures ANOVA with Sidak's post hoc test. **b** A schematic of the experimental procedure. **c** Confocal immunohistochemical images of coronal brain sections from two representative mice, showing the infection of *Gfral*⁺ AP neurons with an AAV expressing TeLC-GFP (left) or GFP only (right). TeLC tumor group, *n* = 9, GFP tumor group, *n* = 8, GFP sham group, *n* = 6. **d** Normalized cumulative food (left) and water (right) intake of the mice before being euthanized (TeLC tumor, *n* = 9, GFP tumor, *n* = 8, GFP sham, *n* = 6; food, F(2,180) = 98.25, P = 10⁻¹⁵; from day −4 to 0: TeLC tumor vs. GFP tumor, P = 0.00004, 9.48 × 10⁻⁹, 1.36 × 10⁻¹¹, 10⁻¹⁵, and P = 10⁻¹⁵, respectively; GFP tumor vs. GFP sham, P = 0.0016, 1.66 × 10⁻⁵, 7.7 × 10⁻⁹, 10⁻¹⁵, and 10⁻¹⁵, respectively; water, F(2180) = 2.889, P = 0.06; two-way repeated-measures ANOVA with Sidak's post hoc test). **e** Inguinal fat (left) and quadriceps muscle (right) mass at 23 days after tumor inoculation (TeLC tumor, *n* = 9, GFP tumor, *n* = 8, GFP sham, *n* = 6; fat, F = 28.74, P = 1.3 × 10⁻⁶, ****P = 0.00007, ****P = 10⁻⁶; muscle, F = 9.324, P = 0.0014, **P = 0.0015, *P = 0.014; one-way ANOVA followed by Tukey's multiple comparisons test). Data in a, d, e are presented as mean ± s.e.m. Source data are provided as a Source Data file.

It's noteworthy that suppressing *Il6ra* in AP neurons (Fig. 6f) appeared less effective than anti-IL-6 antibody i.c.v. infusion in reducing weight loss (Figs. 4e and 5b). This could be because that viral infection can only target a fraction of the relevant neurons. In addition, *Il6ra* in other cell types (such as glia cells) of the AP, or *Il6ra* in other brain areas (e.g., the ME, which experiences elevated IL-6 during late stages of caner progression when cachexia has started (Supplementary Fig. 3a)), may contribute to cachectic symptoms as well.

In the majority of our experiments, we euthanized the different groups of animals at the same timepoints, and noticed that the control groups and experimental groups had no differences in their tumor mass (Figs. 5e, 7g, 8c and 10c), indicating that our manipulations do not affect tumor growth. The differences in tumor mass between the control groups and experimental groups in other experiments (Supplementary Figs. 4e, 7d and 8i) can be explained by the differences in survival time between groups (i.e., the experimental group survived longer than the control group in each case).

One limitation of the current study is that we only used male mice. Cachexia progression in males and females is known to be different, with cachectic phenotypes being usually more severe in males than females[95,96]. This is likely because of the differences in sex hormones, body composition, insulin sensitivity, glucose and lipid metabolism, and energy utilization between sexes. Therefore, effective treatments for cancer cachexia should be sex specific. Whether or not our approach is effective in female mice needs to be investigated in the future.

## Methods
### Mice
All experimental procedures were approved by the Institutional Animal Care and Use Committee of Cold Spring Harbor Laboratory and performed in accordance with the US National Institutes of Health guidelines. Male mice aged 2–4 months were used in all the experiments. Before the experiments, mice were housed under a 12-h light/dark cycle (7 a.m. to 7 p.m. light) in groups of 2–5 animals, with a room temperature (RT) of 22 °C and humidity of 50%. Food and water were available *ad libitum*. Regular chow (PicoLab rodent diet 20, 5053*; physiological value, 3.43 kcal g⁻¹) were provided to all the animals during feeding experiments. When measuring food and water intake and after the surgery (i.e., canulation or tumor inoculation), mice were singly housed in the metabolic cages (see below the "Measuring bodyweight, food intake, and water intake" section).

The Balbc mice (strain number: 000651) and C57BL/6 J (strain number: 000664) were purchased from Jackson laboratory. The *Gfral-p2a-Cre* mice was generated by Stephen Liberles[51].

### Immunohistochemistry
Immunohistochemistry experiments were performed following standard procedures. Briefly, mice were anesthetized with Euthasol (0.2 ml; Virbac, Fort Worth, Texas, USA) and transcardially perfused with 30 ml of PBS, followed by 30 ml of 4% paraformaldehyde (PFA) in PBS. Brains were extracted and further fixed in 4% PFA overnight followed by cryoprotection in a 30% PBS-buffered sucrose solution for 36 h at 4 °C. Coronal sections (50 μm in thickness) were cut using a freezing microtome (Leica SM 2010R). Brain sections were first washed in PBS (3 × 5 min), incubated in PBST (0.3% Triton X-100 in PBS) for 30 min at RT and then washed with PBS (3 × 5 min). Next, sections were blocked in 5% normal goat serum in PBST for 30 min at RT and then incubated with the primary antibody overnight at 4 °C. Sections were washed with PBS (5 × 15 min) and incubated with the fluorescent secondary antibody at RT for 2 h. After washing with PBS (5 × 15 min), sections were mounted onto slides with Fluoromount-G (eBioscience, San Diego, California, USA). Images were taken using an LSM 780 laser-scanning confocal microscope (Carl Zeiss, Oberkochen, Germany).

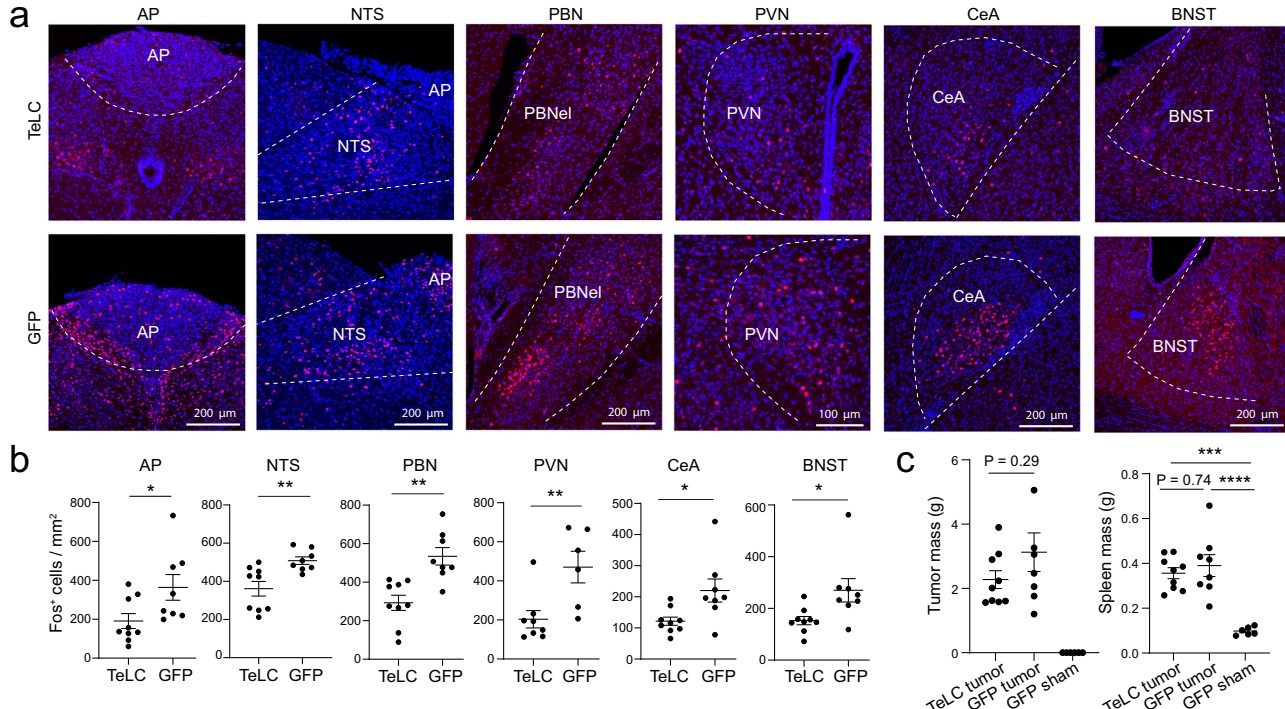

**Fig. 10 | Inhibition of *Gfral⁺* AP neurons attenuates AP network hyperactivity in the Lewis lung cancer (LLC) model. a** Confocal immunohistochemical images showing Fos expression in different brain areas in the mice where *Gfral⁺* AP neurons were infected with the AAV expressing TeLC (top) or GFP (bottom). **b** Quantification of Fos⁺ cells in different brain areas (TeLC group, n = 9 mice, GFP group, n = 8 mice; AP, t = 2.33, *P = 0.0343; NTS, t = 3.3, **P = 0.0049; PBN, t = 4.04, **P = 0.0011; PVN, t = 3.09, **P = 0.0094; CeA, t = 2.62, *P = 0.0193; BNST, t = 2.56, *P = 0.022; t test with false discovery rate adjusted). **c** Tumor (left) and spleen

(right) mass of the mice at 23 days after tumor inoculation (TeLC tumor group, n = 9, GFP tumor group, n = 8, GFP sham group, n = 6; tumor, F = 13.57, P = 0.0002; spleen, F = 19.12, P = 0.00002, ***P = 0.00012, ****P = 0.000035; one-way ANOVA followed by Tukey's multiple comparisons test). AP, area postrema; NTS, nucleus tractus solitarii; PBN, parabrachial nucleus; CeA, central amygdala; PVN, paraventricular nucleus of hypothalamus; BNST, bed nucleus of the stria terminalis. Data in **b**, **c** are presented as mean ± s.e.m. Source data are provided as a Source Data file.

The primary antibodies and dilutions used in this study were: rabbit anti-Fos (1:500, Santa Cruz, sc-52; 1:1000, Cell Signaling Technology, #2250), mouse anti-FLAG (1:1000, Sigma-Aldrich, F1804), rabbit anti-NeuN (1:200; Abcam, ab177487), rabbit anti-mCherry (1:1,000; Abcam, ab167453, GR3213077-3). The fluorophore-conjugated secondary antibodies and dilutions used were Alexa Fluor 488 goat anti-rabbit IgG (H + L; 1:500; A-11008, Invitrogen), Alexa Fluor 647 goat anti-rabbit IgG (H + L; 1:500; A-21244, Invitrogen), Alexa Fluor 594 goat anti-mouse IgG (H + L; 1:500; A-11005, Invitrogen).

### Retro-orbital injection of exogenous IL-6 and its detection in the brain

Biotinylated human IL-6 solution (Acrobiosystems, IL-6-H8218; 2 μg/ml dissolved in saline) was injected into either healthy Balbc mice or cachectic Balbc mice (100 μl per mouse) via retro-orbital injection. For retro-orbital injection, briefly, the animal was anaesthetized with 2% isoflurane. A 27-gauge needle on a 0.5 mL insulin syringe was used for the injection. The animal was placed on its side on a heat pad. The gauge needle was inserted at approximately a 30-45° angle to the eye, lateral to the medial canthus, through the conjunctival membrane. There is a bit of resistance that causes the eye to retreat back into the sinus until the needle pierces through the conjunctiva. The needle was positioned behind the globe of the eye in the retro-bulbar sinus. The biotinylated human IL-6 solution was injected slowly into the retro-bulbar sinus. After the injection, the needle was removed gently and the animal was returned to homecage for recovery. 3 hours after the injection, the animals were euthanized and transcardially perfused with 30 ml of PBS, followed by 30 ml of 4% paraformaldehyde (PFA) in

PBS. Brains were extracted and further fixed in 4% PFA overnight followed by cryoprotection in a 30% PBS-buffered sucrose solution for 36 h at 4 °C. Coronal sections (50 μm in thickness) were cut using a freezing microtome (Leica SM 2010R). Brain sections were incubated in Streptavidin solution (1:1000, ThermoFisher, Alexa Fluor™ 647 conjugate, dissolved in 0.3% PBST) in room temperature for 2 h. After washing with PBS (5 × 15 min), sections were mounted onto slides with Fluoromount-G (eBioscience). Images were taken using an LSM 780 laser-scanning confocal microscope (Carl Zeiss). Mean fluorescence intensity quantification was performed with ImageJ and normalized to the mean fluorescence intensity of images from the saline group.

### Fluorescence in situ hybridization

Single molecule fluorescent in situ hybridization (smFISH) (RNAscope, ACDBio) was used to detect the expression of *Glp1r*, *Il6ra*, *Fos*, *mCherry* and *Gfral* mRNAs in the area postrema. For tissue preparation, mice were first anesthetized under isoflurane and then decapitated. Their brain tissue was first embedded in cryomolds (Sakura Finetek, Catalog number 4566) filled with M-1 Embedding Matrix (Thermo Scientific, Catalog number 1310) then quickly fresh-frozen on dry ice. The tissue was stored at −80 °C until it was sectioned with a cryostat. Cryostat-cut sections (16-μm thick) containing the entire area postrema were collected through the rostro-caudal axis in a series of four slides, and quickly stored at −80 °C until processed. Hybridization was carried out using the RNAscope kit (ACDBio). On the day of the experiment, frozen sections were postfixed in 4% PFA in RNA-free PBS (hereafter referred to as PBS) at RT for 15 min, then washed in PBS, dehydrated using increasing concentrations of ethanol in water (50%, once; 70%, once; 100%, twice; 5 min each). Sections were then dried at RT and incubated

with Protease IV for 30 min at RT. Sections were washed in PBS three times (5 min each) at RT, then hybridized. Probes against *Glp1r* (Catalog number 418851-C3, dilution 1:50), *Il6ra* (Catalog number 438931-O1, dilution 1:50), *Fos* (Catalog number 316921-C2, dilution 1:50), *mCherry* (Catalog number 431201-C3, 1;50), and *Gfral* (Catalog number 417021-C2, dilution 1:50) were applied to the area postrema sections. Hybridization was carried out for 2 h at 40 °C. After that, sections were washed twice in 1× Wash Buffer (Catalog number 310091; 2 min each) at RT, then incubated with the amplification reagents for three consecutive rounds (30 min, 15 min and 30 min, at 40 °C). After each amplification step, sections were washed twice in 1× Wash Buffer (2 min each) at RT. Finally, fluorescence detection was carried out for 15 min at 40 °C. Sections were then washed twice in 1× Wash Buffer (2 min each), incubated with DAPI for 2 min, washed twice in 1× Wash Buffer (2 min each), then mounted with a coverslip using mounting medium. Images were acquired using an LSM780 confocal microscope equipped with 20x and 40x lenses, and visualized and processed using ImageJ and Adobe Illustrator. Cell counting and mean fluorescence intensity quantification of the images were performed with ImageJ.

To determine whether a cell was double or triple labeled, different fluorescent channels (DAPI, green, red, far-red) of multi-channel images were visualized individually using ImageJ, in which the channels of interest were merged together. Transcript dots were manually assigned as somatic localization of RNA transcripts according to their colocalization with DAPI nuclear staining. Thresholds were adjusted based on the expression levels of each probe to minimize false-positive cells. For example, we used a minimum of 3 dots for *Il6ra*, 10 dots for *Gfral*, and 25 dots for *Glp1r*. If a DAPI-positive nucleus is surrounded by the dots of two different transcripts, then it is counted as a double-labeled cell. If a DAPI-positive nucleus is surrounded by the dots of three different transcript, then it is counted as a triple-labeled cell.

### Single guide RNA (sgRNA) design and lentiviral production for CRISPR/dCas9 interference

sgRNAs targeting the *Il6ra* transcription start site (TSS) were designed using CHOPCHOP[97]. Seven *Il6ra* sgRNAs (sgRNA-1 to sgRNA-7) as well as a sgRNA targeting the *lacZ* promoter (*LacZ* sgRNA) were cloned into the Lenti U6-sgRNA/Ef1a-mCherry plasmid (Addgene #114199), as described previously[98,99]. The eight sgRNA plasmids, Lenti SYN-FLAG-dCas9-KRAB-MeCP2 plasmid (Addgene #155365), and the two helper plasmids pCMV-VSV-G (Addgene #8454) and psPAX2 (Addgene #12260) were purified with the NucleoBond Xtra Midi EF kit (Takara 740420). *Il6ra* knockdown efficiency was assessed by transient transfection of sgRNA and dCas9-KRAB-MeCP2 into the mHypoA hypothalamic neural cell line (Cedarlane Labs, clone clu-175). 3 μg of each sgRNA plasmid and 3 μg of dCas9-KRAB-MeCP2 plasmid were co-nucleofected into $1 \times 10^6$ mHypoA cells resuspended in a 1:1 mixture of Ingenio Electroporation reagent (Mirus Bio 50111) and OptiMEM (Gibco 31985062), using program A-033 on the Nucleofector 2b (Lonza). The cells were harvested 60 hours post-nucleofection. DAPI- and mCherry-positive cells were collected by FACS. The *Il6ra* mRNA was extracted and the knockdown efficiency was measured by RT-qPCR. The two most effective sgRNAs, sgRNA-4 (−23 to −41 of TSS) and sgRNA-6 (−163 to −182 of TSS), resulting in 67% and 35% knockdown of *Il6ra* expression in mHypoA cells, respectively, were used for in vivo experiments. FLAG-dCas9-KRAB-MeCP2, *Il6ra* sgRNA-4, *Il6ra* sgRNA-6, and *lacZ* sgRNA lentiviruses were produced in HEK293T cells. Lentiviral pellets were resuspended in 30 uL DPBS, aliquoted and flash-frozen on dry ice, and stored at −80 °C. Physical and functional titers (Supplementary Table 1) were obtained using the Lenti-X qRT-PCR Titration Kit (Takara 631235) and qPCR of genomic DNA following HEK293T transduction[100], respectively.

**Plasmids for lentiviral production.** lenti SYN-FLAG-dCas9-KRAB-MeCP2 was a gift from Jeremy Day (Addgene plasmid # 155365; http://n2t.net/addgene:155365; RRID:Addgene_155365).

lenti U6-sgRNA/EF1a-mCherry was a gift from Jeremy Day (Addgene plasmid # 114199; http://n2t.net/addgene:114199; RRID:Addgene_114199).

pCMV-VSV-G was a gift from Bob Weinberg (Addgene plasmid # 8454; http://n2t.net/addgene:8454; RRID:Addgene_8454).

psPAX2 was a gift from Didier Trono (Addgene plasmid # 12260; http://n2t.net/addgene:12260; RRID:Addgene_12260).

### Adeno-associated viruses (AAVs)

The AAV-CMV-DIO-EGFP-2A-TeLC vector was a gift from Dr. Wei Xu at UT Southwestern. A custom virus (AAV-DJ) based on this vector was produced by WZ Biosciences Inc (Rockville, MD 20855). pAAV-hSyn-DIO-EGFP was purchased from Addgene (Watertown, MA 02472, USA). All viruses were aliquoted and stored at −80 °C until use.

### Stereotaxic surgery

All surgery was performed under aseptic conditions and body temperature was maintained with a heating pad. Standard surgical procedures were used for stereotaxic injection. Briefly, mice were anesthetized with isoflurane (3% at the beginning and 1% for the rest of the surgical procedure), and were positioned in a stereotaxic injection frame and on top of a heating pad maintained at 35 °C. A digital mouse brain atlas was linked to the injection frame to guide the identification and targeting of different brain areas (Angle Two Stereotaxic System, http://myNeuroLab.com). We used the following coordinates for injections in the area postrema: −7.65 mm from bregma, 0 mm lateral from the midline, and 4.7 mm vertical from the skull surface. For virus injection, we made a small cranial window (1–2 mm²) for each mouse, through which a glass micropipette (tip diameter, ~5 μm) containing viral solution was lowered down to the target. For AAVs, about 0.3 μl of viral solution was injected. For lentiviruses, 0.2-0.3 μl of viral mixture (the dCas9 and the sgRNA viruses were mixed at a volume:volume ratio of 2:1) was injected. Viral solutions were delivered with pressure applications (5–20 psi, 5–20 ms at 1 Hz) controlled by a Picospritzer III (General Valve) and a pulse generator (Agilent). The speed of injection was ~0.1 μl/10 min. We waited for at least 10 min following the injection before slowly removing the injection pipette. After injection, the incision was sealed by surgical sutures and the animal was returned to homecage for recovery.

### Colon-26 (C26) adenocarcinoma cells

C26 cells were cultured in complete growth medium consisting of RPMI 1640 medium with Glutamine (#11-875-093; Thermo Fisher) containing 10% of heat-inactivated Fetal Bovine Serum (FBS) (#10-438-026; Thermo Fisher) and 1x Penicillin-Streptomycin solution (#15-140-122; Thermo Fisher) under sterile conditions. 1x Trypsin-EDTA (#15400054; Thermo Fisher) was used for cell dissociation. Cells were resuspended in FBS-free RPMI and viable cells were counted using a Vi-Cell counter prior to subcutaneous injection of $2 \times 10^6$ viable cells diluted in 100 μL RPMI into the right flank of each BALB/c mouse.

### FC1245 cells

The FC1245 clonogenic cell line[63,64] used in the pancreatic ductal adenocarcinoma (PDAC) model was generously provided by David Tuveson at Cold Spring Harbor Laboratory. FC1245 cells were cultured in Dulbecco's Modified Eagle's Medium (DMEM) (#10-013-CV; Corning) complete growth medium, containing 10% of heat-inactivated FBS (#10-438-026; Thermo Fisher) and 1x Penicillin-Streptomycin solution (#15-140-122; Thermo Fisher) under sterile conditions. All cell lines were routinely tested and confirmed negative for mycoplasma contamination. Under isoflurane anesthesia, a small surgical incision was made in the upper-left quadrant of the abdomen. The skin layers, fascia, and muscle wall were reflected to expose the pancreas. The tail of the pancreas was injected with either one million cancer cells suspended in 40 μl of PBS or an equal volume of cell-free PBS (as control).

After the inoculation, the abdominal wall and fascia layers were sutured, followed by two surgical skin clips to close the incision site.

### Lewis lung carcinoma (LLC) cells

LL/2 (LLC1) cells were obtained from ATCC (American Type Culture Collection; #CRL-1642) and cultured in Dulbecco's Modified Eagle's Medium (DMEM) (#30-2002; ATCC) complete growth medium, with 10% of heat-inactivated FBS (#10-438-026; Thermo Fisher) and 1x Penicillin-Streptomycin solution (#15-140-122; Thermo Fisher) under sterile conditions. 1x Trypsin-EDTA (#15400054; Thermo Fisher) was used for cell dissociation. Cells were resuspended in FBS-free DMEM and viable cells were counted using a Vi-Cell counter prior to sub-cutaneous injection of. $2 \times 10^6$ viable cells were diluted in 100 μL FBS-free DMEM and were subcutaneous injected into the right flank of each C57BL/6 mouse.

The humane endpoints were weight loss of 15% of bodyweight and tumor burden of 20 mm in diameter for subcutaneous tumors. For the C26 tumor model, as soon as a mouse was detected to lose equal to or more than 15% of bodyweight, it would be euthanized. Of note, by the time of detection, some of the mice might have already lost more than 20% of their bodyweight, mainly because of the fact that they lost bodyweight so quickly (e.g., some mice lost more than 10% of their bodyweight in one day). To minimize the harm to the animals, all the animals were closely monitored on a daily basis. Animals which reached the humane endpoints were euthanized immediately. For the PDAC model, the animals were euthanized according to the timepoint used in previous studies[63,64]. For the LLC model, the mice were euthanized once the tumor reached maximum size (i.e., 20 mm in diameter).

### Intracerebroventricular (i.c.v.) infusion of anti-IL-6 antibody

For the experiments in Fig. 5 where the i.c.v. infusion of anti-IL-6 antibody occurred before tumor inoculation, an osmotic device of 200 uL volume and a release rate of 0.25 uL/hour consisting of a cannula, connecting line, metal flow moderator and pump (#AP-2004; Alzet) was placed in a subcutaneous pocket and stereotactically implanted into the right lateral ventricle of Balbc mice. The coordinates for targeting the lateral ventricle are -0.5 mm from bregma, 1.25 mm lateral from the midline, and 2.5 mm vertical from the skull surface. Prior to use, the infusion device was assembled and equilibrated in saline for at least 40 hours. The pump was filled with either the InVivoMAb rat anti-mouse IL-6 (clone MP5-20F3, #BE0046; BioXCell) or an InVivoMAb rat IgG1 isotype control (anti-HRPN, #BE0088; BioXCell). Both antibodies were diluted in PBS to achieve continuous infusion at a 10 mg/mL dose. The release of the antibody lasts for 28 days. After the pump implantation, the C26 tumor was inoculated in the subcutaneous space of the mice immediately. When a mouse in the isotype control group developed cachexia, we would randomly choose a mouse in the anti-IL-6 antibody group and euthanize both animals at the same time. This ensured that anti-IL-6 antibody group and isotype control group were euthanized at the same timepoints.

For the experiments in Fig. 4 & Supplementary Fig. 4 where the i.c.v. infusion of anti-IL-6 antibody occurred after tumor inoculation and before the onset of cachexia, on day 10 or 12 post-C26 injection, an osmotic device of 200 uL volume and a release rate of 0.5 uL/hour consisting of a cannula, connecting line, metal flow moderator and pump (#AP-2001; Alzet) was placed in a subcutaneous pocket and stereotactically implanted into the right lateral ventricle of the C26-tumor bearing BALB/c mice for a period of 14 days. Prior to use, the infusion device was assembled and equilibrated in saline overnight. The pump was filled with either the InVivoMAb rat anti-mouse IL-6 (clone MP5-20F3, #BE0046; BioXCell) or an InVivoMAb rat IgG1 isotype control (anti-HRPN, #BE0088; BioXCell). Both antibodies were diluted in PBS to achieve continuous infusion of a 5 mg/mL dose. Pump replacement surgery was performed after 14 days. The isotype control

group started with 6 mice (1 received the surgery on day 10, 5 received the surgery on day 12), while the anti-IL6 group started with 10 (5 received the surgery on day 10, 5 received the surgery on day 12). One of the animals in the anti-IL-6 group was found accidentally dead after canula implantation. For this mouse, we only collected the fat and muscle tissue data but not the Fos data (since we could not do the perfusion needed for Fos staining).

### Measuring bodyweight, food intake, and water intake

Food and water intake monitoring cages (BioDAQ Unplugged, Research Diets, Inc., New Brunswick, NJ 08901 USA) were used to measure the food intake and water intake of the animals. Mice were singly housed in these cages. Food and water were placed in an extended hopper which can be reached by the animal. The bodyweight of the animal, weight of the food and water in the hopper were measured daily at 4 pm. The cachectic mice which lost >15% of bodyweight were euthanized and the tissues were collected for further analysis. Blood glucose concentrations were measured from whole venous blood using an automatic glucose monitor (Bayer HealthCare Ascensia Contour).

### CSF extraction

Mice were anesthetized using isoflurane and placed on a stereotactic alignment instrument (Kopf Instruments). A 2-cm incision was made over the cisterna magna and the trapezius and paraspinal muscles were reflected. Blood and extracellular fluid lying over the cisterna magna were carefully removed to avoid CSF contamination. A glass micro-pipette (tip diameter of ~400–800 μm) was stereotactically inserted into the cisterna magna for capillary action-based CSF collection.

### Blood and plasma measurements

Blood glucose concentrations were measured from whole venous blood using an automatic glucose monitor (Bayer HealthCare Ascensia Contour). Tail vein bleeding was performed using a scalpel via tail venesection without restraint. Blood samples were collected from tail bleed using heparin-coated hematocrit capillary tubes to avoid coagulation. Samples were then centrifuged at 19,283 g for 5 min at 4 °C. Plasma was collected in a new tube, snap frozen in liquid nitrogen and stored at -80 °C. IL-6 and GDF-15 levels were measured in plasma using the mouse IL-6 Quantikine ELISA Kit (#M6000B; R&D Systems) and the Mouse/Rat GDF-15 Quantikine ELISA Kit (#MGD150; R&D) respectively.

### Brain tissue lysis and IL-6 quantification

Mice were transcardially perfused with saline and the area postrema was collected, snap frozen in liquid nitrogen and stored at -80 °C until further analysis. Tissue was placed into 2-mL round-bottom homogenizer tubes pre-loaded with Stainless Steel beads (#69989; Qiagen) and filled up with lysis buffer (#AA-LYS-16ml; RayBiotech) supplemented with Protease Inhibitor Cocktail (#AA-PI; Raybiotech) and Phosphatase Inhibitor Cocktail Set I (#AA-PHI-I; RayBiotech). Samples were homogenized in Tissue Lyser II (#85300; Qiagen) for 5 minutes and then lysates were centrifuged at 4 °C for 20 minutes at 22,136 g. The supernatant was harvested and kept on ice if testing fresh or sored at -80 °C. The Bicinchoninic Acid (BCA) Method was used to determine protein concentration in lysates. IL-6 levels were quantified in the lysates using a Mouse IL-6 ELISA specific for lysates (#ELM-IL6-CL-1; RayBiotech).

### In vitro electrophysiology

Acute slices were obtained from two- to three-month-old mice. Mice were anaesthetized with isoflurane (4%) before rapid decapitation. The brain was rapidly removed, and coronal slices (300 μm) containing the AP were cut using a HM650 Vibrating-blade Microtome (Thermo Fisher Scientific). Slices were cut in ice-cold dissection buffer (110.0 mM

Choline chloride, 25.0 mM $NaHCO_3$, 1.25 mM $NaH_2PO_4$, 2.5 mM KCl, 25.0 mM glucose, 0.5 mM $CaCl_2$, 7.0 mM $MgCl_2$, 11.6 mM ascorbic acid, and 3.1 mM pyruvic acid, and bubbled with 95% $O_2$ and 5% $CO_2$) and subsequently transferred to a recovery chamber containing artificial cerebrospinal fluid (ACSF) solution (containing 118 mM NaCl, 2.5 mM KCl, 26.2 mM $NaHCO_3$, 1 mM $NaH_2PO_4$, 20 mM Glucose, 2 mM $CaCl_2$ and 2 mM $MgCl_2$, pH 7.4, and saturated with 95% $O_2$ and 5% $CO_2$) at 34 °C. The slices were maintained at 34 °C for at least 40 minutes and subsequently at room temperature (20-24 °C). Recordings were made in a continuously flow of ACSF and bubbled with 95% $O_2$/5% $CO_2$.

Whole-cell patch-clamp recordings were obtained with Multi-clamp 700B amplifiers and pCLAMP 10 software (Molecular Devices; Sunnyvale, California, USA) and was guided using an Olympus BX51 Microscope equipped with both transmitted and epifluorescence light sources (Olympus Corporation, Shinjuku, Tokyo, Japan).

Synaptic responses were recorded at holding potentials of -70mV (for AMPA receptor-mediated responses), and 0 mV (for GABAA receptor-mediated responses) and were low-pass filtered at 1 kHz. The internal solution for voltage-clamp experiments contained 115 mM Cesium methanesulfonate, 20 mM CsCl, 10 mM HEPES, 2.5 mM $MgCl_2$, 4 mM $Na_2$-ATP, 0.4 mM $Na_3$-GTP, 10 mM Na-phosphocreatine, and 0.6 mM EGTA, pH 7.2. Miniature EPSCs were recorded in the presence of tetrodotoxin (1 μM) and picrotoxin (100 μM). Spontaneous IPSCs were recorded in the presence of AP-5 (100 μM) and CNQX (5 μM). The EPSCs and IPSCs were analyzed using Mini Analysis software (Synaptosoft).

## Statistics & reproducibility

All statistics are indicated where used. Statistical analyzes were conducted using GraphPad Prism version 6.0 (GraphPad Software, Inc., La Jolla, CA). Statistical comparisons were performed using Student's t test or ANOVA. All comparisons were two tailed. Statistic hypothesis testing was conducted at a significance level of 0.05. To ensure that results are reproducible, we included detailed methods, sources of reagents and protocols for all the experiments. All experiments were successfully repeated with the indicated numbers of mice. No statistical method was used to predetermine sample size. After histological inspection, the mice in which viral injection was mis-targeted were excluded. No other data were excluded from the analyzes. All mice were randomly assigned to different groups. The experiment and analysis in Supplementary Fig. 4 were performed blind. In other experiments, the investigators were not blinded to allocation during experiments and outcome assessment.

## Reporting summary

Further information on research design is available in the Nature Portfolio Reporting Summary linked to this article.

## Data availability

The authors declare that the data supporting the findings of this study are available within the paper, its supplementary information files and source data. Source data are provided with this paper. Correspondence and requests for materials should be addressed to B.L. (bli@cshl.edu, libo@westlake.edu.cn). Source data are provided with this paper.

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

## Acknowledgements

We thank Dr. Stephen D. Liberles for sharing the *Gfral-p2a-Cre* mice. We thank Dr. Roberto Malinow and Dr. Chuchu Zhang for comments on earlier versions of the manuscript. We thank Radhashree Sharma for technical assistance and members of the Li laboratory for helpful discussions. B.G. is supported by a fellowship from the National Institutes of Health (NIH) (F31MH124365). This work was performed in collaboration with the Cold Spring Harbor Laboratory (CSHL) shared resources, which are supported by NIH (Cancer Center Support Grant 5P30CA045508: Animal, Animal and Tissue Imaging Shared Resources). The authors are supported by the Lustgarten Foundation, where D.A.T. is a distinguished scholar and Director of the Lustgarten Foundation–designated Laboratory of Pancreatic Cancer Research. D.A.T. is also supported by the Thompson Foundation, the Pershing Square Foundation, the CSHL and Northwell Health Affiliation, the Northwell Health Tissue Donation Program, the Cold Spring Harbor Laboratory Association, the National Institutes of Health (5P30CA45508, U01CA210240, R01CA229699, U01CA224013, 1R01CA188134, and 1R01CA190092), and a gift from the Simons Foundation (552716). Y.P. is supported by the National Cancer Institute (R50CA211506). J.T. is supported by NIH (R01MH113628). T.J. is supported by funding from Cancer Grand Challenges (NIH: 1OT2CA278690-01; CRUK: CGCATF-2021/100019), the Mark Foundation for Cancer Research (20-028-EDV), the Osprey Foundation, Fortune Footwear, CSHL, and developmental funds from CSHL Cancer Center Support Grant (5P30CA045508). B.L. is supported by NIH (R01MH101214, R01MH108924, R01NS104944, R01DA050374), the CSHL and Northwell Health Affiliation, and the Key R&D Program of Zhejiang (2024SSYS0031).

## Author contributions

Q.S., D.V.D.L., M.F., B.G, M.W., J.T., T.J., and B.L. designed research; Q.S., D.V.D.L., M.F., B.G, and M.W. performed research; Q.S., D.V.D.L., M.F., B.G, M.W., J.T., T.J. and B.L. analyzed data; Y.P. and D.A.T. assisted with the FC1245 cancer model; and Q.S. and B.L. wrote the paper with inputs from all authors.

## Competing interests

Q.S., D.V.D.L. and B.L. are listed as inventors on a patent application concerning the development of new strategies for treating cancer cachexia. All other authors declare no competing interests.
