## [Peer Review File · Nature Communications]

Reviewers' Comments:

Reviewer #1:

Remarks to the Author:

The manuscript by Sun et al. demonstrates that circulating IL-6 binds to the area postrema in the murine hindbrain, where it activates AP neurons by binding to the IL-6R. They go on to demonstrate that neutralization of IL-6 in the brain improves cachexia and extends lifespan in murine cachexia models. This is associated with decreases in activation of AP neurons. They then demonstrate that deletion of IL-6R in the AP, as well as silencing of GFRAL-expressing neurons also ameliorate symptoms of cachexia in the murine cancer models.

There are several points that should be addressed:

Major Points:

- 1) Expression of IL6R has been documented in the rodent brain in several prior publications, and is found outside of the AP by many groups. Furthermore, expression of IL6R is increased in the hypothalamus by stress (e.g., Aniszewska et al, 2015), and in brain endothelium and CVOs by a variety of inflammatory insults (e.g., Vallieres et al, 2002). Given that cancer is a chronic inflammatory state, it would make more sense to evaluate biotinylated IL-6 binding in animals with established systemic inflammation (including the cancer models used here) than at baseline. Similarly, IL6 itself is strongly induced in the CNS of animals with systemic inflammation, including in CVOs and parenchymal brain structures, which in turn would greatly increase the exposure of various brain regions to IL6 synthesized on the inside of the BBB.
- 2) The data presented in figure 2d-f are misleading as presented. The majority of the animals in the isotype group died within a 48-hour period after a major surgical procedure (subcutaneous cannula implantation, intracerebroventricular cannulation), that caused acute weight loss (and likely relative anorexia and hypodipsia) in the majority of the animals. This has a major impact when the data are plotted as "days to sacrifice". Furthermore, the body weight in figure 2e is normalized to day -5 for all animals, but the animals in the IL6 antibody group had survived for much longer, and had larger tumors (some more than 1.5g), which are included in the weight gain curves (the same problem occurs in figure 4h). It is also unclear why animals underwent the second procedure "10 or 12" days after the tumor inoculation. Given that the majority of animals die between day 10 and 13, this is not a trivial issue. How was the interval (10 vs. 12) days chosen, and how many animals in each treatment group were in each category? The fact that the isotype group were all seriously hypoglycemic at the time that they were killed suggests that they were near death at that time. This impacts the cFos data shown in figure 2g. Is it really reasonable to study cFos expression, and attribute effects specifically to central IL6 activity, in animals that are in such vastly different systemic physiological states? Overall, a more relevant study design would start the anti-IL6 infusion much earlier relative to the tumor inoculation, to avoid the acute impact of CNS surgery. Similarly, cFos analysis really needs to be done on the same experimental day for both groups.
- 3) It appears that the quantification of GFRAL cells expressing IL6ra (fig 4d) was done at the time that each animal was killed. Given that these time points are nearly a month different from each other in the two comparison groups, time becomes a confounding factor in the interpretation of these data. It would be much more convincing to demonstrate this change in cohorts analyzed at a fixed time point after lentivirus injection (e.g., in the animals from figure s5).
- 4) Without a control group, it is impossible to know if the LLC tumor is inducing cachexia. It is concerning that mice were allowed to carry tumors of ~5-6 grams.

Minor Points:

- 1) There is no description of how a cell was determined to be double (or triple) labeled in figure 1.
- 2) The data in figure S1 is used to support the idea that systemic IL6 only binds to the AP. Are there similar data for cFos induction (as shown in the AP for figure 1). Similarly, are there data for IL6 levels in other CVOs in the C26 animals? If not, this is worth discussing as a weakness. Please provide the primary data for figure 2b, which will allow the reader to understand if the apparent increase in IL6 is driven by a decrease in total protein.
- 3) The NIH has specifically requested that investigators stop using the word "sacrifice" to describe the killing of experimental mice.
- 4) The isotype control group in figure 2 started with 10 animals, but cFos analysis was done on just six of those animals. How was this down-selection accomplished?

- 5) Food intake should be normalized to body weight wherever possible.
- 6) There is no actual evidence that the anti-IL6 antibody lowered circulating or CSF IL6. Indeed, there is extensive overlap between groups in both measures. This should be discussed.
- 7) The anti-flag staining shown in figure 4b is sufficiently overexposed that it is really impossible to discern if there is true double-labeling with mCherry. What is the evidence that this expression is limited to neurons?
- 8) Figure 4 gives the appearance that there are fewer GFRAL neurons in the IL6ra group than in the lacZ group. In fact, the same figure appears to show a lower density of cells overall (DAPI staining). Was this actually evaluated? It is also unclear why there are so many fewer infected cells (mCherry stain) between groups.
- 9) The majority of these studies do not measure muscle mass (or fat mass), which are defining features of cachexia. It is not clear why this was not done.
- 10) The cachexia in the LLC model is best described as both milder, and far more variable than the c26 model. Also, it is known that the c26 model is a relative outlier amongst most murine cancer models in the systemic release of IL6 (e.g., figure 5a vs. figure s2a). These points may be worth discussion.

Reviewer #2:

Remarks to the Author:

Cancer cachexia (weight loss and anorexia) significantly hinders the survival of cancer patients and studies are needed for identifying therapeutic strategies that ameliorate this syndrome. Sun et al present a nice series of studies testing the hypothesis that IL-6 cytokine signaling in area postrema (AP) neurons significantly contributes to cancer cachexia (weight loss and anorexia). Using a mouse model, their initial studies show that the AP exhibits higher aggregation of systemically administered IL-6 compared to other brain areas, suggesting that the AP is a potential entry point for IL-6 to induce cancer cachexia. Consistent with the idea that IL-6 signaling in the brain contributes to cancer cachexia, the authors show that injecting an IL-6 antibody into the lateral ventricle prevents the onset of cancer cachexia in an implantable tumor model; this treatment also attenuates cancer-induced activation of AP neurons and brain areas that are downstream of the AP. As a direct test of whether IL-6 signaling in AP neurons underlies cancer cachexia, they knock down IL-6 receptor in the AP using virally delivered CRISPR constructs and show that this treatment attenuates cancer-induced cachexia. Further implicating the AP in cancer cachexia, they show that inactivation of AP Gfral (GDF-15 cytokine receptor) neurons also attenuates cancer cachexia. Thus, the authors provide several contributions to the research field. I am enthusiastic about the manuscript but have a few reservations:

1. In addition to the AP, past research has also implicated the median eminence (another circumventricular organ) and inflammation-induced cachexia. Is systemically administered IL-6 detected here? This seems relevant because IL-6 antibody injection into the lateral ventricle would also affect this hypothalamic area.
2. CRISPR knock down of IL-6 receptor in the area postrema (Fig. 4h) does not appear as effective in preventing weight loss compared to IL-6 antibody injection into the lateral ventricle (Fig. 3e). Does this mean that IL-6 might be acting elsewhere in addition to the AP? At least some discussion is warranted, and if no additional support is provided, reconsideration of the manuscript title to hedge the claim of AP being the mediator of IL-6 related cancer cachexia.
3. Zoomed out images that better show the NTS (for IL-6 detection, RNAscope studies, Fos staining, and CRISPR knock down of IL-6 receptor) would be appreciated. This nucleus lies below the AP and also expresses IL-6 and GDF-15 receptors, thus showing this area is important for interpreting the results (which claim that the AP and not NTS is critical for cancer cachexia).
4. Methods regarding the endpoint for tumor studies are lacking. What were the criteria for ending a study and were they predetermined? Were mice actually allowed to succumb to the disease or were they euthanized after losing a certain amount of weight? Answers to these questions are needed for interpreting the survival experiments.

5. The authors consistently describe their treatments as “ameliorating” cancer cachexia. Amelioration implies that something bad was made better. But their manipulations (IL-6 antibody, CRISPR KO, and neuron inactivation) all began before the onset of cachexia, therefore should be described as “attenuating” symptoms. While this may appear as a nitpicky distinction, it raises an important question that has clinical implications: do any of the manipulations reverse (i.e., ameliorate) cachexia if applied after onset of the syndrome?

6. The tumor experiments should have been conducted with a control group that was not implanted with tumor cells. This is especially important for interpreting the extent to which treatments attenuate anorexia. Yes, there are significant differences between tumor bearing animals that receive control versus experimental treatment, but is the food intake of the experimental group made normal or is it still depressed compared to animals that do not have cancer?

Reviewer #3:

Remarks to the Author:

These are fascinating results showing a quite extraordinary effect on tumor growth and cachexia via manipulation of IL6 signaling in area postrema neurons in mouse models. As they say, extraordinary claims require extraordinary evidence, which is somewhat lacking here.

The manuscript uses outdated models of cancer cachexia - C26 and LLC, neither of which reflects human disease or is credible in cancer research. The results should be reproduced in an orthotopic or genetic model of cancer that is well established in the field.

As well, all key experiments should be repeated, preferably independently by other operators, given that many are small in sample size. When repeating the studies, there should be attention to blinding and randomization and reporting of whether mice under the same manipulation were co-housed or whether mice were randomized to cages so the results are not a cage effect.

Tumor mass should be reported in the main figure and considered in the interpretation of results and discussion. Food intake graphs are not clear given very small error bars when they represent 7-8 mice housed 2-5 per cage. Perhaps just sum food intake across groups instead of trying to average an underpowered study design.

Limitations including that of sex should be discussed in the manuscript.

Dear referees,

Thank you for the helpful suggestions. For the revision, we have conducted a number of new experiments to address your concerns, and also modified the paper according to your suggestions. The major new experiments are:

1. We examined whether circulating IL-6 can enter the area postrema (AP) and other brain areas in cachectic mice (Fig. 1b-d; Extended Data Fig. 1) in addition to healthy mice.
2. We examined Fos expression in the AP and other brain areas following systemic application of IL-6 in both healthy control mice and cachectic mice (Fig. 1b, e, f; Extended Data Fig. 2).
3. We did anti-IL-6 antibody i.c.v. infusion before tumor inoculation, which prevented cachexia (Fig. 3), in addition to the experiment where the infusion was performed after tumor inoculation and just before the onset of cachexia (current Extended Data Fig. 5 & 6).
4. We have repeated the major results in a new orthotopic pancreatic tumor (FC1245 PDAC) model (Fig. 5 and Extended Data Fig. 12).
5. We measured IL-6 levels in the cortex and the median eminence (ME), which is another circumventricular organ, during cancer progression (Extended Data Fig. 4).
6. We conducted new single molecule fluorescent *in situ* hybridization (smFISH) experiments and immunohistochemistry experiments (Extended Data Fig. 3; Extended Data Fig. 8).

We have also revised some of the figures and made modifications in the paper based on your suggestions and additional new results. The major changes are colored in red in the main text and Methods.

We feel that our paper is much improved by adding the new experiments and by revising the paper. We sincerely appreciate your thoughtful suggestions.

Below, please find our point-by-point responses to your comments.

Sincerely,
Bo Li

Point-by-point responses to referees' comments (in bold):

Reviewers' comments:

Reviewer #1 - Cancer cachexia therapy, mouse models (Remarks to the Author):

The manuscript by Sun et al. demonstrates that circulating IL-6 binds to the area postrema in the murine hindbrain, where it activates AP neurons by binding to the IL-6R. They go on to demonstrate that neutralization of IL-6 in the brain improves cachexia and extends lifespan in murine cachexia models. This is associated with decreases in activation of AP neurons. They then demonstrate that deletion of IL-6R in the AP, as well as silencing of GFRAL-expressing neurons also ameliorate symptoms of cachexia in the murine cancer models.

There are several points that should be addressed:

Major Points:

1) Expression of IL6R has been documented in the rodent brain in several prior publications, and is found outside of the AP by many groups. Furthermore, expression of IL6R is increased in the hypothalamus by stress (e.g., Aniszewska et al, 2015), and in brain endothelium and CVOs by a variety of inflammatory insults (e.g., Vallieres et al, 2002). Given that cancer is a chronic inflammatory state, it would make more sense to evaluate biotinylated IL-6 binding in animals with established systemic inflammation (including the cancer models used here) than at baseline. Similarly, IL6 itself is strongly induced in the CNS of animals with systemic inflammation, including in CVOs and parenchymal brain structures, which in turn would greatly increase the exposure of various brain regions to IL6 synthesized on the inside of the BBB.

We thank the referee for the helpful suggestions. We have performed additional experiments to evaluate biotinylated IL-6 binding in the brain of tumor-bearing mice that had developed cachexia (new Fig. 1b-d and Extended Data Fig. 1). The new results show that, in both baseline and cancer cachexia conditions, circulating IL-6 binds only to the area postrema (AP) but no other brain areas. Interestingly, the binding in the AP under cachectic condition is lower than that in baseline (Fig. 1c and d), likely because that endogenous IL-6 in the AP increases during cachexia and competes with the exogenous IL-6 for interacting with IL-6 receptors.

Regarding whether systemic inflammation induced by cancer can generally increase IL6 in various brain areas, we have performed new experiments which show that this is not the case. While we found that IL-6 in the AP increased starting from day 7 after tumor inoculation (new Fig. 2b, left; note that we have now substantially increased the number of mice in this experiment), we did not detect any increase in IL-6 levels in the median eminence (ME, another circumventricular organ suggested by Referee #2) until the mice had already developed cachexia (new Extended Data Fig. 4a), which typically occurs at ~2 weeks after the tumor inoculation. In addition, we did not detect any increase in IL-6 levels in the cortex throughout the experiment (new Extended Data Fig. 4b). These results suggest that under our experimental conditions, peripheral tumor does not lead to a general increase in IL-6 levels in the brain.

2) The data presented in figure 2d-f (**it should be the original Figure 3d-f**) are misleading as presented. The majority of the animals in the isotype group died within a 48-hour period after a major surgical procedure (subcutaneous cannula implantation, intracerebroventricular cannulation), that caused acute weight loss (and likely relative anorexia and hypodipsia) in the majority of the animals. This has a major impact when the data are plotted as “days to sacrifice”. Furthermore, the body weight in figure 2e is normalized to day -5 for all animals, but the animals in the IL6 antibody group had survived for much longer, and had larger tumors (some more than 1.5g), which are included in the weight gain curves (the same problem occurs in figure 4h).

To address the referee’s concerns, we added new experiments in which we performed the antibody infusion before tumor inoculation, and euthanized all mice at the same time after tumor inoculation to make sure that both groups had comparable tumor sizes (new Figure

3). These experiments produced similar results to the old antibody infusion experiments (current Extended Data Fig. 5 & 6).

In addition, we would like to clarify that in the old antibody infusion experiments, we performed the same surgical procedures (subcutaneous cannula implantation, intracerebroventricular cannulation) at the same time point for both the isotype group and the anti-IL-6 group, but the anti-IL-6 group survived much longer than the isotype group despite the surgery (see current Extended Data Fig. 5c). Indeed, 9 out of 10 mice in the anti-IL-6 group recovered from the surgery and survived without cachexia, whereas 5 out of 6 in the isotype group could not recover and became cachexic, and therefore had to be euthanized. This result indicates that anti-IL-6 can prevent cachexia.

Furthermore, in the experiments where we used CRISPR/dCas9 system to suppress the expression of IL-6R in the AP (current Figure 4 and Extended Data Figure 11), the surgical procedure (viral injection) was performed much earlier, 14 days before the inoculation of tumor. Also, to address the issue of longer survival leads to larger tumors, in additional experiments we had euthanized all mice at the same time after tumor inoculation, and therefore both groups had comparable tumor sizes (current Extended Data Figure 10). In all these experiments, we observed similar effects, that is, suppressing IL-6R expression in AP neurons reduced body weight loss and tissue loss associated with cancer cachexia.

It is also unclear why animals underwent the second procedure “10 or 12” days after the tumor inoculation. Given that the majority of animals die between day 10 and 13, this is not a trivial issue. How was the interval (10 vs. 12) days chosen, and how many animals in each treatment group were in each category?

The reason we performed canula implantation at 10 or 12 days after tumor inoculation is because in the C26 mouse model, most of the animals start to develop cachexia within this time window. We chose to infuse the antibody when animals were at the onset of cachexia. We reasoned that, if we were able to rescue the animals at this late stage, we would most likely also be able to protect the animals from getting cachexia if the intervention were performed at earlier stages.

Animals were chosen randomly to receive the cannulation surgery on either 10 or 12 days after tumor inoculation, and were assigned randomly to either the isotype control group or the anti-IL-6 group. The isotype control group started with 6 animals (1 received the surgery on day 10, 5 received the surgery on day 12), while the anti-IL6 group started with 10 animals (5 received the surgery on day 10, 5 received surgery on day 12). One of the animals in the anti-IL-6 group was found accidentally dead after canula implantation. For this mouse, we only collected the fat and muscle tissue data but not the Fos data (since we could not do the perfusion needed for Fos staining). We have now provided this information in the Methods, which can be found in lines 836-840.

The fact that the isotype group were all seriously hypoglycemic at the time that they were killed suggests that they were near death at that time. This impacts the cFos data shown in figure 2g (should be the original Figure 3g). Is it really reasonable to study cFos expression, and attribute

effects specifically to central IL6 activity, in animals that are in such vastly different systemic physiological states?

In this particular experiment (current Extended Data Figure 5g, h), we wanted to use Fos expression as a readout of how cancer progression (including the cachectic state) is associated with specific patterns of brain activation, and how our manipulations might affect the activation patterns. Such patterns initially may depend on IL-6, especially at the early stage of cancer progression, but later may reflect network activity in the brain and the systemic physiological states of the animals, as the referee pointed out. However, by suppressing IL-6 or IL-6R in the brain we can normalize brain activity patterns and improve animals' systemic physiological states. Therefore, IL-6 signaling is critical but is only one part of the many changes in cancer cachexia.

Besides the experiment in Extended Data Figure 5 (g, h), we have results from multiple Fos experiments (including newly added experiments) that support our hypotheses: (1) Fos expression 3 hours after IL6 retro-orbital injection in healthy mice (new results in current Fig. 1e, f; Extended Data Fig. 2). (2) Fos expression before the onset of cachexia, 11 days after tumor implantation (current Figure 2c, d). (3) Fos expression measured at the same timepoints after tumor inoculation for both the experimental groups and the control groups (current Figure 3 (new results) and Extended Data Figure 10). In these experiments, similar patterns of Fos expression were observed following IL-6 injection or at different stages of cancer progression, and were normalized by suppressing IL-6 or IL-6R.

We have now clarified this point in the Discussion, which could be found in lines 297-305.

Overall, a more relevant study design would start the anti-IL6 infusion much earlier relative to the tumor inoculation, to avoid the acute impact of CNS surgery. Similarly, cFos analysis really needs to be done on the same experimental day for both groups.

We thank the referee for the helpful suggestions. As mentioned above, we have added new experiments in which we performed the antibody infusion before tumor inoculation, and examined Fos expression on the same experimental day for both groups (new Figure 3). In addition, similar results were obtained from other experiments where Fos analysis was done on the same experimental day for both groups (Figure 2c, d; Extended Data Figure 10).

3) It appears that the quantification of GFRAL cells expressing IL6ra (fig 4d) was done at the time that each animal was killed. Given that these time points are nearly a month different from each other in the two comparison groups, time becomes a confounding factor in the interpretation of these data. It would be much more convincing to demonstrate this change in cohorts analyzed at a fixed time point after lentivirus injection (e.g., in the animals from figure s5).

We thank the referee for the helpful suggestion. We have now added a new figure in which we analyzed IL6ra expression at a fixed time point after lentivirus injection for both groups (see new Extended Data Fig. 8c, d).

4) Without a control group, it is impossible to know if the LLC tumor is inducing cachexia. It is concerning that mice were allowed to carry tumors of ~5-6 grams.

We have now added a control group (“GFP sham”) as the referee suggested. The results can be found in new Fig. 6d, e.

As for the tumor size, there were only 2 mice with tumors over 4 grams. We followed strict experimental procedures, which were approved by the Institutional Animal Care and Use Committee of Cold Spring Harbor Laboratory and performed in accordance with the US National Institutes of Health guidelines. The criteria for end-stage mice are the following: Experiments will be completed before tumor development or tumor-associated diseases causing death or a significant deterioration in the mice. Mice bearing C26, LLC or FC1245 tumors will be weighed daily. Once tumor bearing mice lose >15% body weight, they will be euthanized. The maximum tumor size for subcutaneous tumor is 20 mm in diameter in a mouse. Once a tumor reaches this size the mouse will be euthanized. Mice in chronic pain or distress that cannot be relieved by analgesics will be euthanized.

We have now added this information to the Methods, which can be found in lines 803-807.

Minor Points:

1) There is no description of how a cell was determined to be double (or triple) labeled in figure 1.

We have now added the description in Methods, which can be found in lines 702-710.

2) The data in figure S1 is used to support the idea that systemic IL6 only binds to the AP. Are there similar data for cFos induction (as shown in the AP for figure 1). Similarly, are there data for IL6 levels in other CVOs in the C26 animals? If not, this is worth discussing as a weakness.

We have now added Fos induction data in other brain areas after IL-6 injection (current Extended Data Fig. 2. In addition, we have now added new data for IL-6 levels in other brain areas, including another CVO (ME, see Extended Data Fig. 1 and new Extended Data Fig. 4; also see the above response to main point #1).

Please provide the primary data for figure 2b, which will allow the reader to understand if the apparent increase in IL6 is driven by a decrease in total protein.

We have now provided the primary data (see new Figure 2b, right).

3) The NIH has specifically requested that investigators stop using the word “sacrifice” to describe the killing of experimental mice.

We have now changed the term to “euthanize” throughout the paper.

4) The isotype control group in figure 2 started with 10 animals, but cFos analysis was done on just six of those animals. How was this down-selection accomplished?

This actually is not the case. The isotype control group started with 6 animals, while the anti-IL-6 group started with 10 animals. As mentioned above (in response to main point #2) one of the animals in the anti-IL-6 group was found accidentally dead after canula implantation. For this mouse, we only collected the fat and muscle tissue data but not the Fos data (since we could not do the perfusion needed for Fos staining). However, there was no “down-selection” for either the isotype control group or the anti-IL-6 group.

We apologize for this confusion. We have now provided this information in the Methods (see lines 836-840).

5) Food intake should be normalized to body weight wherever possible.

We have now normalized food and water intake to body weight throughout the paper.

6) There is no actual evidence that the anti-IL6 antibody lowered circulating or CSF IL6. Indeed, there is extensive overlap between groups in both measures. This should be discussed.

i.c.v. infusion of the anti-IL-6 antibody did not change IL-6 levels in the plasma (current Extended Data Fig. 6d, left), but had a tendency to decrease IL-6 in the CSF ($P = 0.067$, current Extended Data Fig. 6d, right). We have now discussed this in the text (see lines 159-161).

7) The anti-flag staining shown in figure 4b is sufficiently overexposed that it is really impossible to discern if there is true double-labeling with mCherry. What is the evidence that this expression is limited to neurons?

We thank the referee for raising this issue. We have now conducted experiments on additional mice and taken images in which we avoided overexposure (new Fig. 4b; new Extended Data Fig. 8a, b). We also did a double staining against FLAG and NeuN to verify that the expression is specific to neurons (see new Extended Data Fig. 8a, b).

8) Figure 4 gives the appearance that there are fewer GFRAL neurons in the IL6ra group than in the lacZ group. In fact, the same figure appears to show a lower density of cells overall (DAPI staining). Was this actually evaluated? It is also unclear why there are so many fewer infected cells (mCherry stain) between groups.

After checking the images carefully, we found that the magnifications of the images were different between the two groups. The images for the two groups were also not from the same location within the area postrema. We believe that these issues have caused the confusion. We sincerely apologize for this. To address this issue and the issue in main point #3 (see above), we have performed new experiments, based on which we made a new figure (see new Extended Data Fig. 8c, d).

9) The majority of these studies do not measure muscle mass (or fat mass), which are defining features of cachexia. It is not clear why this was not done.

We actually have measured muscle mass and fat mass in ALL the experiments – see Fig. 3e, Fig. 5c, Fig. 6e, Extended Data Fig. 6a & b; Extended Data Fig. 9b & c, Extended Data Fig. 10c & d, Extended Data Fig. 11g & h, and Extended Data Fig. 12d.

10) The cachexia in the LLC model is best described as both milder, and far more variable than the c26 model. Also, it is known that the c26 model is a relative outlier amongst most murine cancer models in the systemic release of IL6 (e.g., figure 5a vs. figure s2a). These points may be worth discussion.

We have now discussed these points (see lines 287-295).

Reviewer #2 - Neuro sensing of appetite (Remarks to the Author):

Cancer cachexia (weight loss and anorexia) significantly hinders the survival of cancer patients and studies are needed for identifying therapeutic strategies that ameliorate this syndrome. Sun et al present a nice series of studies testing the hypothesis that IL-6 cytokine signaling in area postrema (AP) neurons significantly contributes to cancer cachexia (weight loss and anorexia). Using a mouse model, their initial studies show that the AP exhibits higher aggregation of systemically administered IL-6 compared to other brain areas, suggesting that the AP is a potential entry point for IL-6 to induce cancer cachexia. Consistent with the idea that IL-6 signaling in the brain contributes to cancer cachexia, the authors show that injecting an IL-6 antibody into the lateral ventricle prevents the onset of cancer cachexia in an implantable tumor model; this treatment also attenuates cancer-induced activation of AP neurons and brain areas that are downstream of the AP. As a direct test of whether IL-6 signaling in AP neurons underlies cancer cachexia, they knock down IL-6 receptor in the AP using virally delivered CRISPR constructs and show that this treatment attenuates cancer-induced cachexia. Further implicating the AP in cancer cachexia, they show that inactivation of AP Gfral (GDF-15 cytokine receptor) neurons also attenuates cancer cachexia. Thus, the authors provide several contributions to the research field. I am enthusiastic about the manuscript but have a few reservations:

We thank the referee for the positive comments.

1. In addition to the AP, past research has also implicated the median eminence (another circumventricular organ) and inflammation-induced cachexia. Is systemically administered IL-6 detected here? This seems relevant because IL-6 antibody injection into the lateral ventricle would also affect this hypothalamic area.

We thank the referee for this suggestion. We have now checked biotinylated IL-6 levels in the median eminence (ME) after its systemic administration (new Extended Data Fig. 1). We didn't find obvious binding of the exogenous IL-6 in this area under either baseline or cachectic state. Furthermore, after the IL-6 systemic administration, the ME didn't show increased Fos expression (see new Extended Data Fig. 2).

2. CRISPR knock down of IL-6 receptor in the area postrema (Fig. 4h) does not appear as effective in preventing weight loss compared to IL-6 antibody injection into the lateral ventricle (Fig. 3e). Does this mean that IL-6 might be acting elsewhere in addition to the AP? At least some discussion is warranted, and if no additional support is provided, reconsideration of the manuscript title to hedge the claim of AP being the mediator of IL-6 related cancer cachexia.

We thank the referee for bringing up this point. It is possible that the viral infection in Figure 4 did not effectively cover all relevant neurons. It is also possible that additional cell types or brain areas are involved in mediating the effects of anti-IL-6 infusion, which are not targeted by our viral infection. We have now taken this into account in the discussion (see lines 342-347).

3. Zoomed out images that better show the NTS (for IL-6 detection, RNAscope studies, Fos staining, and CRISPR knock down of IL-6 receptor) would be appreciated. This nucleus lies below the AP and also expresses IL-6 and GDF-15 receptors, thus showing this area is important for interpreting the results (which claim that the AP and not NTS is critical for cancer cachexia).

We have now added images containing the NTS in different experiments (see current Extended Data Figures 1, 2, 3, 5g & h, 8, 10i & j, and 12e & f; and current main Figures 2c & d, 3f & g, 4b, 5d & e, and 6f & g. Of note, NTS belongs to the “AP network” that we define in this study, and participates in the responses to cancer cachexia. We have added more discussions to this point (see lines 297-313).

4. Methods regarding the endpoint for tumor studies are lacking. What were the criteria for ending a study and were they predetermined? Were mice actually allowed to succumb to the disease or were they euthanized after losing a certain amount of weight? Answers to these questions are needed for interpreting the survival experiments.

We followed strict experimental procedures, which were approved by the Institutional Animal Care and Use Committee of Cold Spring Harbor Laboratory and performed in accordance with the US National Institutes of Health guidelines. The criteria for the endpoint of tumor studies are the following: Experiments will be completed before tumor development or tumor-associated diseases causing death or a significant deterioration in the mice. Mice bearing C26, LLC or FC1245 tumors will be weighed daily. Once tumor bearing mice lose >15% body weight, they will be euthanized. The maximum tumor size for subcutaneous tumor is 20 mm in diameter in a mouse. Once a tumor reaches this size the mouse will be euthanized. Mice in chronic pain or distress that cannot be relieved by analgesics will be euthanized.

We have now added this information to the paper (see lines 803-807).

5. The authors consistently describe their treatments as “ameliorating” cancer cachexia. Amelioration implies that something bad was made better. But their manipulations (IL-6 antibody, CRISPR KO, and neuron inactivation) all began before the onset of cachexia, therefore should be described as “attenuating” symptoms. While this may appear as a nitpicky distinction,

it raises an important question that has clinical implications: do any of the manipulations reverse (i.e., ameliorate) cachexia if applied after onset of the syndrome?

We thank the referee for raising this issue. We have now changed the term “ameliorating” to “attenuating” throughout the paper, as suggested.

6. The tumor experiments should have been conducted with a control group that was not implanted with tumor cells. This is especially important for interpreting the extent to which treatments attenuate anorexia. Yes, there are significant differences between tumor bearing animals that receive control versus experimental treatment, but is the food intake of the experimental group made normal or is it still depressed compared to animals that do not have cancer?

We thank the referee for the suggestion. We have now included food and water intake data (and other related data) of normal animals without tumor (see new Fig. 6d, e, h and Extended Data Fig. 12).

Reviewer #3 - Cancer cachexia, IL-6, mouse models (Remarks to the Author):

These are fascinating results showing a quite extraordinary effect on tumor growth and cachexia via manipulation of IL6 signaling in area postrema neurons in mouse models.

We thank the referee for the positive comments.

As they say, extraordinary claims require extraordinary evidence, which is somewhat lacking here.

The manuscript uses outdated models of cancer cachexia - C26 and LLC, neither of which reflects human disease or is credible in cancer research. The results should be reproduced in an orthotopic or genetic model of cancer that is well established in the field.

We have now reproduced the major results in an orthotopic pancreatic ductal adenocarcinoma (PDAC) model (see new Fig. 5 and Extended Data Fig. 12).

As well, all key experiments should be repeated, preferably independently by other operators, given that many are small in sample size. When repeating the studies, there should be attention to blinding and randomization and reporting of whether mice under the same manipulation were co-housed or whether mice were randomized to cages so the results are not a cage effect.

We thank the referee for these suggestions. One experiment (the original Fig. 2b) was underpowered, but now we have increased the number of mice for this experiment (see the current Fig. 2b). As detailed in the Reporting Summary, we have made sure that all experiments were sufficiently repeated, the manipulations were conducted under the exact same conditions for different groups, and all mice were randomly assigned to different groups. The experiment and analysis in Extended Data Fig. 6 were performed blind.

Although other data collection and analysis were not performed blind to the conditions of the experiments, equal parameters and processes were applied for all groups. Further, all behavioral experiments were performed under consistent conditions and the related data were collected and analyzed in an unbiased way. Regarding the comment on potential cage effects, all the animals were single-housed when we measured the metabolic parameters, as we described in the Methods (see lines 619-621 and lines 842-850). Therefore, there should not be any cage effect.

Tumor mass should be reported in the main figure and considered in the interpretation of results and discussion. Food intake graphs are not clear given very small error bars when they represent 7-8 mice housed 2-5 per cage. Perhaps just sum food intake across groups instead of trying to average an underpowered study design.

We have now reported the major tumor mass results in main Fig. 3e, 5c, and 6h. Additional tumor mass results can be found in Extended Data Fig. 6e, 9d, 10g and 11i. We have also added discussions about the tumor mass as suggested (see lines 349-355).

As to the food intake graphs, all the animals were single-housed, that is, they were not housed 2-5 per cage. We apologize that this information was not obvious in the initial submission. It was described in the Methods section “Measuring bodyweight, food intake, and water intake” (see lines 842-850). We have now also added the information at the beginning of Methods (lines 619-621).

It is indeed notable that the food intake of individual animals was very consistent across different days, with very little variability. However, this is typical of such innate behavior across many studies (Borner et al., 2020; Campos et al., 2017; Flint et al., 2016; Roman et al., 2016), which is much less variable than learned behaviors of animals.

Limitations including that of sex should be discussed in the manuscript.

We have now discussed this issue as suggested (see lines 357-362).

References:

Borner, T., Shaulson, E.D., Ghidewon, M.Y., Barnett, A.B., Horn, C.C., Doyle, R.P., Grill, H.J., Hayes, M.R., and De Jonghe, B.C. (2020). GDF15 Induces Anorexia through Nausea and Emesis. *Cell Metab* 31, 351-362 e355.

Campos, C.A., Bowen, A.J., Han, S., Wisse, B.E., Palmiter, R.D., and Schwartz, M.W. (2017). Cancer-induced anorexia and malaise are mediated by CGRP neurons in the parabrachial nucleus. *Nat Neurosci* 20, 934-942.

Flint, T.R., Janowitz, T., Connell, C.M., Roberts, E.W., Denton, A.E., Coll, A.P., Jodrell, D.I., and Fearon, D.T. (2016). Tumor-Induced IL-6 Reprograms Host Metabolism to Suppress Anti-tumor Immunity. *Cell Metab* 24, 672-684.

Roman, C.W., Derkach, V.A., and Palmiter, R.D. (2016). Genetically and functionally defined NTS to PBN brain circuits mediating anorexia. *Nat Commun* 7, 11905.

Reviewers' Comments:

Reviewer #2:

Remarks to the Author:

The authors addressed my suggestions. This is a nice collection of studies and I hope it gets published.

Reviewer #3:

Remarks to the Author:

The authors have responded rather thoroughly to the comments and concerns and a stronger manuscript has emerged. This is an interesting and significant contribution to the field.

Dear referees,

Thank you for the positive comments and the support for publication of our paper.

Sincerely,

Bo Li

Reviewers' comments:

Reviewer #2 (Remarks to the Author):

The authors addressed my suggestions. This is a nice collection of studies and I hope it gets published.

Reviewer #3 (Remarks to the Author):

The authors have responded rather thoroughly to the comments and concerns and a stronger manuscript has emerged. This is an interesting and significant contribution to the field.